# Rhythmic interactions between the mediodorsal thalamus and prefrontal cortex precede human visual perception

Benjamin J. Griffiths[1], Tino Zaehle[2], Stefan Repplinger [2,3], Friedhelm C. Schmitt [2], Jürgen Voges[4], Simon Hanslmayr [5] & Tobias Staudigl [1✉]

The thalamus is much more than a simple sensory relay. High-order thalamic nuclei, such as the mediodorsal thalamus, exert a profound influence over animal cognition. However, given the difficulty of directly recording from the thalamus in humans, next-to-nothing is known about thalamic and thalamocortical contributions to human cognition. To address this, we analysed simultaneously-recorded thalamic iEEG and whole-head MEG in six patients (plus MEG recordings from twelve healthy controls) as they completed a visual detection task. We observed that the phase of both ongoing mediodorsal thalamic and prefrontal low-frequency activity was predictive of perceptual performance. Critically however, mediodorsal thalamic activity mediated prefrontal contributions to perceptual performance. These results suggest that it is thalamocortical interactions, rather than cortical activity alone, that is predictive of upcoming perceptual performance and, more generally, highlights the importance of accounting for the thalamus when theorising about cortical contributions to human cognition.

[1] Department of Psychology, Ludwig-Maximilians-Universität München, Munich, Germany. [2] Department of Neurology, Otto-von Guericke-University, Magdeburg, Germany. [3] ESF International Graduate School on Analysis, Imaging and Modelling of Neuronal and Inflammatory Processes, Otto-von-Guericke University, Magdeburg, Germany. [4] Department of Stereotactic Neurosurgery, Otto-von-Guericke University, Magdeburg, Germany. [5] Centre for Cognitive Neuroimaging, Institute for Neuroscience and Psychology, University of Glasgow, Glasgow, UK. ✉email: tobias.staudigl@psy.lmu.de

Thalamic contributions to cognition have been profoundly underestimated[1]. Contrary to a cortico-centric view of cognition[2], a whole host of cognitive phenomena rely on the thalamus and its interactions with the cortex[3,4]. For example, animal models suggest that it is the interactions between the mediodorsal thalamus and the prefrontal cortex, as opposed to the actions of the prefrontal cortex alone, that dictate the outcome of tasks that have traditionally been thought of as "prefrontal-dependent" (e.g. attentional control; working memory[5–8]). This thalamic dependency is not surprising considering that the prefrontal cortex has literally been defined as any frontal region that receives innervation from the mediodorsal thalamus[9,10]. As such, an interaction between these two regions in service of cognition seems plausible, yet evidence for such a phenomenon in humans is conspicuously absent.

It is a challenge to record human thalamic electrophysiological activity directly from the source, and this challenge is compounded by the difficulty to record such activity simultaneously with cortical activity. However, with access to simultaneous iEEG-MEG recordings, we can begin to address the relevance of thalamocortical interactions to human cognition—in this case, with a focus on visual detection. Within the cortex, visual detection has been linked to pre-stimulus prefrontal low-frequency activity (6–14 Hz)[11–16], but, as highlighted above, the prefrontal cortex doesn't act in isolation. One could therefore postulate that these pre-stimulus prefrontal low-frequency rhythms reflect connections to mediodorsal thalamus through so-called thalamocortical loops[17–20]. While such low-frequency loops have been demonstrated in animals, evidence for similar loops in humans is scarce. Therefore, to explore this phenomenon in humans, we analysed simultaneously-recorded intracranial electroencephalography (iEEG; targeting the mediodorsal thalamic nuclei) and whole-brain magnetoencephalography (MEG) in six patients as they completed a visual detection task (see Fig. 1a, b; see methods for commentary on sample size). Additionally, we analysed MEG recordings in twelve healthy participants undergoing the same task.

## Results

In the first instance, we asked whether the phase of ongoing low-frequency activity in the mediodorsal thalamus was predictive of visual detection. Morlet wavelets were used to extract measures of instantaneous phase, and then the phase angles for "hits" (i.e., when the correct stimulus was selected) and "misses" (i.e., when the incorrect stimulus was selected) were contrasted using the phase bifurcation index (PBI)[11]. We expected to find a positive PBI, which would indicate that there is a consistent phase angle difference between the two conditions prior to stimulus onset.

Indeed, using this approach, we observed a positive PBI in the mediodorsal thalamus that was significantly greater than what would be expected by chance (peaking at 7 to 8 Hz, 600 to 300 ms prior to stimulus onset; mean cluster $t(5) = 5.90$, $p_{clus} < 0.001$, Bayes Factor [$BF_{10}$] = 23.49; see Fig. 1c), indicating that there was an optimal mediodorsal thalamic phase for visual detection. This could be observed in every participant (see Fig. 1d, e). No robust phase bifurcation was observed in additional anterior thalamic recordings ($t(4) = 4.20$, $p_{clus} > 0.5$, $BF_{10} = 5.63$; though no difference in PBI was observed between the anterior and mediodorsal thalami: $t(5) = 7.52$, $p_{clus} = 0.094$, $BF_{10} = 25.84$; see Supplementary Fig. 2).

Notably, the phase of the ongoing low frequency activity of several participants seemed to undergo a rapid shift reminiscent of a phase reset following stimulus onset (see Fig. 1d). To investigate this, we looked at how spectral power fluctuated as an interaction between time (pre-stimulus vs. post-stimulus) and

signal derivation technique (total power vs. evoked power). Previous work[21] has suggested that an interaction in which evoked post-stimulus power increases relative to pre-stimulus power, but total power does not, would indicate that the phase of the signal has aligned across trials. However, we observed no such interaction ($F(1, 5) = 1.04$, $p = 0.355$; see Supplementary Fig. 5), suggesting that phase did not reorganise consistently across participants following stimulus onset.

While several studies have linked low-frequency power to visual perception[22–26], we did not observe any significant relationship between mediodorsal thalamic low-frequency power and visual detection ($t(5) = 2.92$, $p_{clus} = 0.453$, $BF_{10} = 2.83$; see Supplementary Fig. 4).

When shifting focus from the thalamus to the cortex, we found that similar pre-stimulus phase patterns within the source-localised medial prefrontal cortex were predictive of upcoming perceptual performance (mean cluster $t(5) = 10.62$, $p_{clus} = 0.016$, $BF_{10} = 198.21$; see Fig. 1f). This effect was replicated in the healthy control sample (mean cluster $t(11) = 3.52$, $p_{clus} = 0.031$, $BF_{10} = 10.76$; see Fig. 1g) with highly similar spatial localisation, and conforms to earlier reports of the phase of low-frequency prefrontal oscillations predicting upcoming perceptual performance[11,12,15]. While there were minor differences in the timing and spectral profile of the pre-stimulus effects in the patient and control samples, this was not significant (mean cluster $t(16) = 3.15$, $p_{clus} = 0.662$; $BF_{10} = 7.71$). There was, however, a strong negative PBI following stimulus onset for the healthy controls relative to the patient sample (mean cluster $t(16) = −6.44$, $p_{clus} < 0.001$, $BF_{10} = 1558.32$; see Fig. 1f, g). This negative PBI seemed to be driven by the evoked response to the stimulus (see Supplementary Fig. 6). We were unable to ascertain why the evoked response effect was restricted solely to the healthy controls, but given that this effect is restricted solely to the post-stimulus window, and no post-stimulus effect could retroactively alter a pre-stimulus effect, we feel that this open question does not undermine our central results.

Previous studies have also observed phase bifurcation over the dorsal attention network (e.g. refs. [12,27]). While the positioning of the electrode wires during the patient MEG recording prevents us from reliably probing these more posterior sources (see Supplementary Fig. 7), the healthy control MEG recordings show analogous results to those which have been reported previously (see Supplementary Fig. 8).

Given the presence of perceptually-relevant phase separation in both the mediodorsal thalamus and the medial prefrontal cortex, we then asked whether these two regions connected on a trial-by-trial basis. To this end, we used inter-site phase clustering (ISPC [i.e., phase-locking value across sites[28], where a value of '0' indicates no clustering and '1' indicates maximal phase clustering]) to quantify the pre-stimulus low-frequency phase consistency between the mediodorsal thalamus and every voxel of the source-reconstructed MEG signal. Across all trials, connectivity was greatest between the mediodorsal thalamus and the ipsilateral medial prefrontal cortex, at approximately 8 Hz, and was significantly greater than expected by chance (mean cluster $t(5) = 19.83$, $p_{clus} < 0.001$, $BF_{10} = 2,218.64$; see Fig. 2a–c; see Supplementary Fig. 8). This effect was substantial in all patients (see Fig. 2d). A link between this corticothalamic connectivity and perceptual performance was inconclusive (mean cluster $t(5) = 5.37$, $p_{clus} = 0.188$, $BF_{10} = 17.13$; see Supplementary Fig. 9).

When assessing the directionality of this connectivity using the Phase Slope Index (PSI)[29] across all trials, the medial prefrontal cortex appeared to lead low-frequency activity in the mediodorsal thalamus to a significantly greater degree than chance (mean cluster $t(5) = 5.33$, $p_{clus} < 0.001$, $BF_{10} = 16.73$). This directed connectivity was predictive of perceptual performance, as

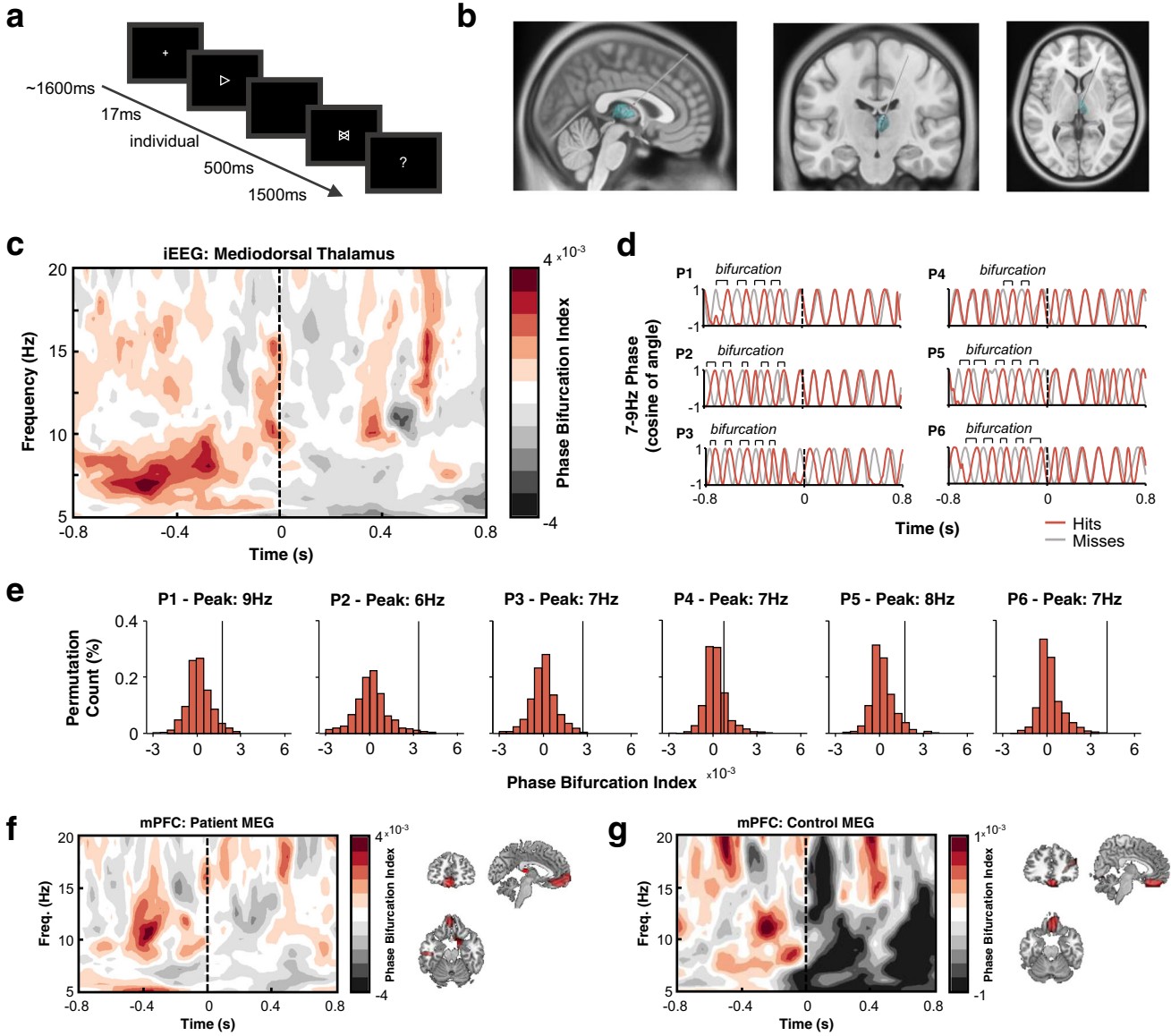

**Fig. 1 Phase bifurcation within the mediodorsal thalamus precedes visual detection. a** Experiment overview. Participants completed a visual detection screen in which an arrow (pointed left or right) was briefly shown before a mask appeared. Participants then indicated which direction they thought the arrow was pointing. **b** Deep brain stimulation electrodes were implanted in the left and right mediodorsal and anterior thalami. See Supplementary Fig. 1 for visualisation of mediodorsal thalamus in the context of other thalamic nuclei. **c** Time-frequency representation depicting mean mediodorsal thalamic phase bifurcation across patients (as measured with iEEG). Higher values indicate greater phase bifurcation (coloured in a deep red; the lowest phase bifurcation values are coloured in dark grey). Time at 0 s represents onset of the target. Substantial low-frequency phase bifurcation was observed prior to stimulus onset. **d** Bandpass-filtered (7–9 Hz) mediodorsal thalamic signal for each participant individually (hits in red; misses in grey). The phases of the two conditions are opposed in all patients. **e** Patient-specific observed phase-bifurcation (black line) compared to a surrogate distribution (red histograms; $n = 1000$ permutations) for individual peak bifurcation frequencies. The comparatively slow frequency effect of participant 2 did not impact the group effect (see Supplementary Fig. 3). **f** MEG-recorded time-frequency representation (left; colour scheme as in **c**) of medial prefrontal phase-bifurcation in patients and source-localisation of the peak of this effect (right; visualised phase bifurcation at −400 ms, 10 Hz; coloured in red; MNI: [−4, 50, −21]). **g** MEG-recorded time-frequency representation of medial prefrontal phase-bifurcation (left; colour scheme as in **c**) in healthy controls and source-localisation of the peak of this effect (right; visualised phase bifurcation at −300 ms, 11 Hz; coloured in red; MNI: [5, 36, −24]). iEEG intracranial electroencephalography, MEG magnetoencephalography, mPFC medial prefrontal cortex. Source data are provided as a Source Data file.

prefrontal-to-thalamic PSI was greater for hits relative to misses (mean cluster $t(5) = 8.26$, $p_{clus} < 0.001$, $BF_{10} = 11.71$; see Fig. 2e, f).

Intriguingly, we also observed directed connectivity in which low-frequency activity in the mediodorsal thalamus preceded low-frequency activity posterior sources (mean cluster $t(5) = -8.15$, $p_{clus} = 0.063$, $BF_{10} = 73.96$). Given that MEG coverage of these posterior sources was inconsistent across

participants (see Supplementary Fig. 7), we have decided to avoid resting any major conclusions based on these thalamus-to-posterior cortex connections. Nonetheless, the interested reader can turn to Supplementary Fig. 10 for more details.

Lastly, we asked whether the mediodorsal thalamus mediates prefrontal contributions to visual detection. To this end, we developed a simple mediation model where prefrontal low-frequency activity could influence perceptual performance

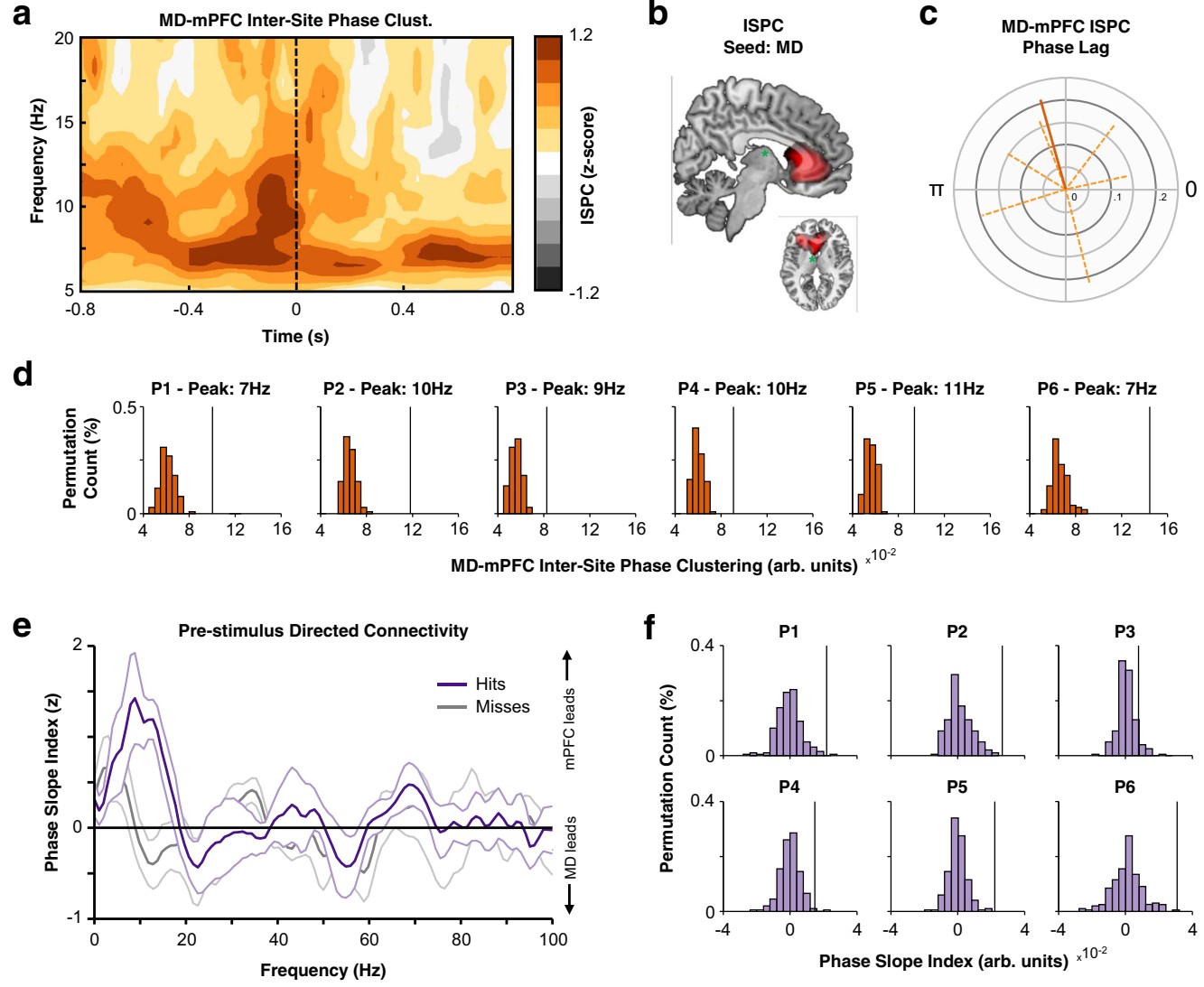

**Fig. 2 Corticothalamic connectivity precedes visual detection. a** Time-frequency representation of phase-based undirected connectivity between intracranial recordings of the mediodorsal thalamus and MEG recordings of the medial prefrontal cortex. Connectivity peaked prior to stimulus onset, at ~8 Hz. Deeper colours of orange indicate high phase-locking, darker greys indicate low phase-locking. **b** Pre-stimulus 8 Hz phase-based undirected connectivity between the mediodorsal thalamus and source-localised MEG signals peak in the ipsilateral prefrontal cortex (phase-locking coloured in red; insert: left reflects ipsilateral hemisphere, right reflects contralateral). Green circle indicates approximate position of mediodorsal thalamic electrode. **c** Polar plot of mean phase lag between mediodorsal thalamus and medial prefrontal cortex. The dark, solid orange line indicates mean phase lag and mean vector length of the participant-specific phase lag angle; light, dotted orange lines indicate mean phase lag and mean vector length per participant). The scale ranges from zero (i.e., no consistent direction) to one (i.e., perfectly consistent lag across participants/trials). Note that the mean phase lag/vector length across participants was calculated only using the phase lags of the individual participants (that is, the calculation was not weighted by participant-specific mean vector length). **d** Patient-specific observed connectivity (black line) compared to surrogate distributions (orange histograms; $n = 1000$ permutations) for individual peak connectivity frequencies. **e** Frequency spectrum for pre-stimulus directed connectivity between medial prefrontal cortex and mediodorsal thalamus (hits in purple, misses in grey; dark line indicates the mean across participants; shaded area indicates the mean $+/-$ the standard error of the mean across participants [$n = 6$ participants]). A positive value indicates that the medial prefrontal cortex leads the mediodorsal thalamus, while a negative value indicates the mediodorsal thalamus leads the medial prefrontal cortex. The medial prefrontal cortex leads the mediodorsal thalamus uniquely for hits. **f** Patient-specific observed directed connectivity (black line) compared to surrogate distributions (purple histograms; $n = 200$ permutations) individual peak directed connectivity frequencies. ISPC Inter-site phase clustering, MD mediodorsal thalamus, mPFC medial prefrontal cortex. Source data are provided as a Source Data file.

directly (see pathway *c'* in Fig. 3a) or indirectly (i.e., via the mediodorsal thalamus; see pathway *ab* in Fig. 3a). In this model, the indirect pathway predicted perceptual performance to a degree greater than what would be expected by chance ($t(5) = 3.85$, $p < 0.001$, $BF_{10} = 12.05$; see Fig. 3b for participant-specific plots of the observed magnitude for the indirect pathway relative to chance). Moreover, when contrasting the magnitude of

pathway *c* (that is: the direct influence of pre-stimulus prefrontal cortical activity on behavioural performance without accounting for thalamic activity) against pathway *c'* (i.e., the direct influence of pre-stimulus prefrontal cortical activity on behavioural performance after accounting for thalamic activity), we found evidence to suggest that the direct influence of pre-stimulus prefrontal cortical activity on behavioural performance was

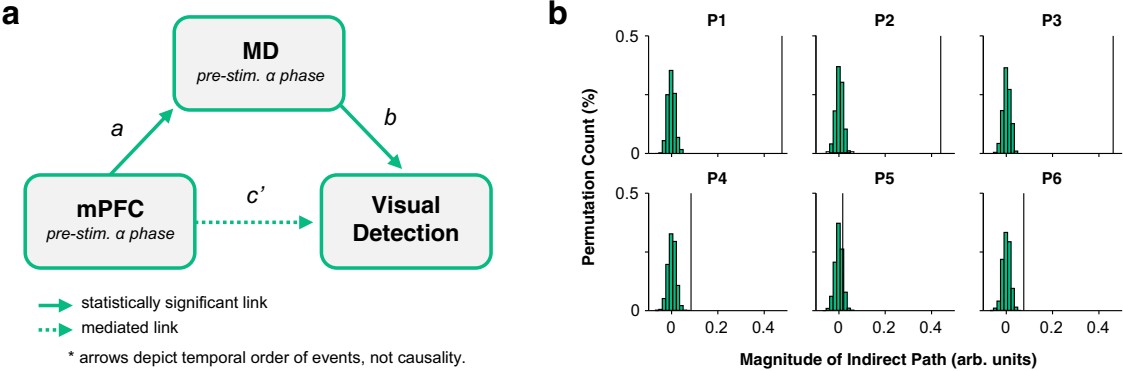

**Fig. 3 Mediodorsal thalamic phase bifurcation mediates prefrontal contributions to visual detection. a** Visualisation of the proposed mediation model. Pre-stimulus low-frequency phase patterns within the medial prefrontal cortex predict visual perceptual performance both directly and/or indirectly via the mediodorsal thalamus. Statistical analysis suggests that the indirect pathway (*ab*) better predicts behavioural performance than the direct pathway. **b** The predictive power of the observed indirect path (*ab*, black line) on behavioural performance relative to chance (green histogram bars; $n = 1000$ permutations). MD mediodorsal thalamus, mPFC medial prefrontal cortex. Source data are provided as a Source Data file.

diminished after accounting for pre-stimulus thalamic activity ($t(5) = 2.26$, $p = 0.031$, $BF_{10} = 3.06$). Similar results can be found when using partial correlations in place of a mediation model (see Supplementary Fig. 11). This suggests that the mediodorsal thalamus mediates prefrontal activity to some degree. However, the direct effect of the medial prefrontal cortex on visual detection continued to explain the outcome to a significant degree after accounting for the indirect effect ($t(5) = 32.99$, $p < 0.001$, $BF_{10} = 33{,}187.91$ [though this is a large Bayes Factor, this is not surprising given the region of interest was selected by identifying the where and when prefrontal rhythms best predicted visual detection prior to accounting for mediodorsal thalamic activity; see methods for details]). This suggests that the prefrontal cortex is not completely redundant in this visual detection task. Nonetheless, these results suggest that mediodorsal thalamic phase bifurcation is not simply an epiphenomenon induced by phase-based correlations with the prefrontal cortex. Rather, the mediodorsal thalamus appears to partially mediate prefrontal contributions to visual perception.

## Discussion

In sum, we find evidence to suggest that visual detection fluctuates as a function of pre-stimulus, low-frequency mediodorsal thalamic phase; a phenomenon which mirrors cortical patterns that have been reported previously (e.g. refs. [11–13]). Moreover, we find that directed coupling between the cortex and thalamus, in which prefrontal activity leads mediodorsal thalamic activity prior to stimulus onset. Critically however, it appears that the mediodorsal thalamus mediates these cortical contributions to visual detection performance (see Fig. 4 for visual summary of the main results).

Of course, a key question remains: what do corticothalamic interactions contribute to visual detection? A recent framework[30] suggests that the thalamus acts as a "Bayesian observer", in which high-order thalamic nuclei use sensory input to update templates of the environment maintained in the cortex[31,32]. Based upon this, one could speculate that the mediodorsal thalamus helps contrast existing cortical templates (maintained in the prefrontal cortex[5,33,34]) with current sensory input. When a mismatch arises between the current input and the prefrontal representation, the mediodorsal thalamus updates this template (e.g., by downweighting the past representation and stabilising the new representation[35]), which is then acted upon[36]. Notably, computational models suggest that these mechanistic interactions

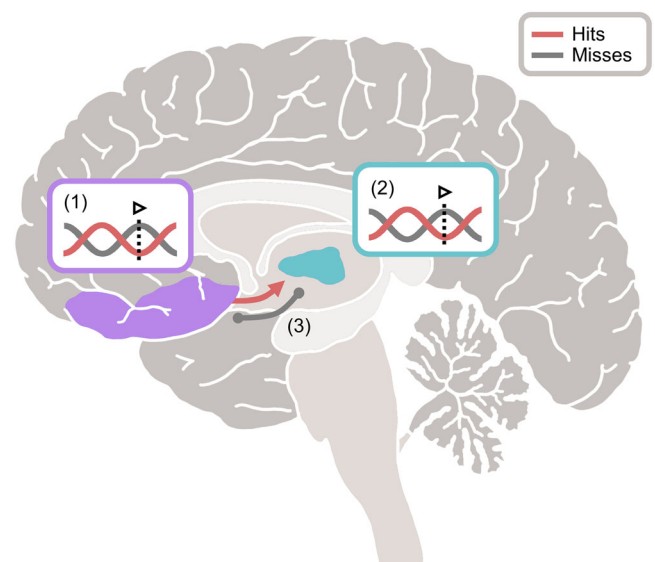

**Fig. 4 Visual depiction of the main findings.** Successful detection of a visual stimulus correlates with several neural phenomena: (1) the stimulus being presented at the optimal, low-frequency phase of ongoing medial prefrontal activity (mPFC in purple; hits in red; misses in grey), (2) the stimulus being presented at the optimal, low-frequency phase of ongoing mediodorsal thalamic activity (mediodorsal thalamus in aqua; hits in red; misses in purple), and (3) directed prefrontal-to-thalamic low-frequency connectivity (hits in red; misses [which displayed undirected connectivity] in grey). Critically, the contribution of the prefrontal cortex to visual detection appears to be mediated by the mediodorsal thalamus.

produce patterns of low-frequency travelling waves between the interacting regions[37], which may explain why corticothalamic connectivity was most prevalent in the low frequencies. If template updating were to breakdown, one could expect that the detection of a transient change in sensory input would fail and corticothalamic low-frequency connectivity would dissipate, which may explain why the directional connectivity from the prefrontal cortex to the mediodorsal thalamus observed here was performance-dependent. While the correlative nature of our data prevents us testing these ideas, future studies which disrupt corticothalamic interactions (e.g., through direct thalamic stimulation) could directly test the causal nature of these hypotheses.

An alternative explanation of the rhythmic corticothalamic interaction stems from works investigating interactions between the pulvinar and cortical attentional networks. Directional interactions between the cortical attentional network and the pulvinar (another high-order thalamic nucleus) rhythmically fluctuate at a rate similar to that which we observe here[20,38]. Functionally speaking, this phase-based switching is thought to correspond to switching between cognitive tasks: namely, sampling the environment and shifting attention[39]. Perhaps a similar phenomenon arises between the prefrontal cortex and mediodorsal thalamus: one phase of the oscillation favours the transfer of sensory/maintained representations to the mediodorsal thalamus, while the other phase supports the updating of the cortical template. This would translate to rhythmic fluctuations in perceptual performance, where stimuli presented during the phase optimal for cortex-to-thalamus communication are more likely to be perceived than those presented during the phase optimal for thalamus-to-cortex communication (which matches with our observation that cortex-to-thalamus directed connectivity is predictive perceptual performance). Again, future studies may turn to methods such as brain stimulation to directly test the causal nature of this hypothesis.

One may be wondering why prefrontal cortical and mediodorsal thalamic phase bifurcation arose at neighbouring, rather than identical, frequencies (~11 Hz and ~8 Hz respectively). While the spectral smearing incurred through the use of wavelets for our measure of inter-site phase clustering and the 6 Hz bandwidth used for the phase-slope index analyses provide a mathematical explanation of connectivity between the two differing frequency bands, it wouldn't explain the physiological underpinnings of such a phenomenon. We speculate, however, that the observed connectivity in conjunction with the mild difference in frequency may relate to travelling waves (e.g. refs. [40,41]); more specifically, travelling waves that come about through weakly-coupled oscillators[42]. Models of weakly-coupled oscillators suggest that travelling waves can couple two regions so long as the oscillator of the transmitting region has a higher intrinsic frequency than the oscillator of the receiving region. In the case of the data presented here, we would anticipate that a travelling wave would begin within the prefrontal cortex (given its higher peak phase bifurcation frequency) and propagate to the mediodorsal thalamus. Notably, such an idea neatly ties to the phase-slope index results which demonstrated directed connectivity from the prefrontal cortex to the mediodorsal thalamus. Moreover, this explanation also aligns with the "Bayesian observer" described above, and the travelling waves inherent in such a hypothesis[37]. Of course, this remains a speculative interpretation of the frequency differences between the two regions as very little is known about corticothalamic travelling waves in humans. Such a hypothesis does, however, present a novel avenue for future research regarding corticothalamic interactions. Future work with more extensive intracranial recordings (a pre-requisite for travelling wave analysis) may provide an answer as to why two regions with differing bifurcating frequencies may relate to a shared phenomenon.

Our observation of low-frequency connectivity between the mediodorsal thalamus and prefrontal cortex suggests that humans exhibit similar thalamocortical loops to those observed in animals[18,20]. To date, studies of these loops in humans are scarce[43], owning to the fact that simultaneous, direct recordings of the specific thalamic nuclei and cortex are rare (see refs. [44–47] for other examples recording from various thalamic nuclei). As such, to understand these moment-by-moment dynamics, the field has had to rely on generalising earlier findings from animal models to humans, rather than studying humans directly. While these models have provided fantastic advances in our understanding of the role of the thalamocortical loops in visual perception, they do have their limitations. Firstly, many of these studies have focused on the pulvinar (e.g., refs. [20,38]), whose anatomical and functional connections to the cortex are notably different to the cortical connections of the mediodorsal thalamus, meaning these results cannot be generalised to explain the role of the mediodorsal thalamus in visual perception. Second, animal models of the prefrontal cortex are limited in their generalisability relative to animal models of other cortical regions owning to the unique evolutionary divergence in structure of the prefrontal cortex[48], meaning prefrontal-thalamic connections in humans remain poorly understood. The data we present here helps overcome these hurdles and demonstrate how synchronised low-frequency activity facilitates interactions between the human cortex and thalamus.

While numerous studies have suggested that prefrontal activity predicts[11–16], and perhaps causes[49–52], fluctuations in perceptual performance, evidence is far from consistent[22,53–56]. Perhaps this is due to overlooking the role of the mediodorsal thalamus and its many connections to the prefrontal cortex. Indeed, given that we found evidence to suggest that the mediodorsal thalamus mediates prefrontal contributions to visual perception, this may explain why cortio-centric investigations of the neural correlates of visual perception produce such inconsistent results.

Beyond the prefrontal cortex, numerous other cortical regions have been shown to engage in visual perceptual processes (e.g., the dorsal attention network[12,27];). Due to the positioning of the iEEG wires in the MEG, however, we were unable to reliably record signals from these regions, and hence investigate how they interact with the mediodorsal thalamus. Despite this however, we observed interesting connectivity dynamics where low-frequency thalamic activity seemingly leads low-frequency activity in the occipital cortex (see Supplementary Fig. 10). In the context of the prefrontal connectivity patterns, one could speculate that signals from the prefrontal cortex pass to the occipital lobe via the mediodorsal thalamus, and may explain why phase opposition effects can be seen across the cortex e.g.,[11,12,27]. Of course, given that these results depend on signals generated from sources with poor MEG sensor coverage, one must take these findings with a grain of salt.

Going forth, our findings emphasise the importance of accounting for the thalamus when probing prefrontal contributions to human cognition[1,30,57], and, more generally, highlight the importance of shifting from a cortico-centric model of human cognition towards a more integrative, thalamocortical model.

## Methods

**Participants**. We recruited six patients (66.6% female, mean age: 41.2 ± 8.9 years, 100% right-handed) with bilateral intracranial depth electrodes implanted in the anterior nuclei of the thalamus for deep brain stimulation therapy of drug-resistant epilepsy for the experiment. We recorded electrophysiological signals from these intracranial electrodes simultaneously with those from an MEG system (see acquisition details overleaf). We obtained informed consent from all participants. The measurements were approved by the Ethics Commission of the Medical Faculty of the Otto-von-Guericke University, Magdeburg.

A sample size of six for an experiment such as this is small (see https://osf.io/tyfwu/ for a constantly-updating table on similar experiments; mean size: 14.8 participants; std: 6.3), though to be expected given the rarity of (i) patients being treated with deep brain stimulation of the thalamus, (ii) access to thalamic electrophysiology in these patients (DBS leads are externalized only in a minority of these patients post-surgery, allowing the present combination of intracranial thalamic recordings and cognitive experiments), and (iii) the summation of the rarity of intracranial recordings and the rarity of the possibility to simultaneously acquire MEG recordings. The problems with such samples are twofold: a heightened likelihood of a false positive, and a heightened likelihood of a false negative. The heightened likelihood of a false positive can, in part, be attributed to the group mean being more easily swayed by a single outlier. To attenuate such a concern here, we have visualised participant-specific effects (see Figs. 1e, 2d, h, 3b) to demonstrate that the effect is not driven by a single participant, but is instead a consistent trend across patients. The heightened likelihood of a false negative can

be attributed to a lack of statistical power. To attenuate this concern, we have supplemented the null-hypothesis testing procedure with a report of Bayes Factor (i.e., the strength of evidence for the alternative, relative to null, hypothesis). While Bayesian analyses are not impervious to issues of low statistical power[58], they can provide a better indication as to whether the absence of an effect is attributable to a genuine null effect, or insufficient power. As a heuristic, a Bayes Factor of less than 3 is considered "anecdotal evidence" for $H_1$ relative to $H_0$, a Bayes Factor between 3 and 10 is considered "moderate evidence" for $H_1$ relative to $H_0$, and a Bayes Factor greater than 10 can be consider "strong evidence" for $H_1$ relative to $H_0$.

We recruited an additional 12 healthy controls (50% female, mean age = 27.6 ± 6.5 years, 100% right-handed), who did not suffer epilepsy and therefore had no intracranial electrodes, to complete the same task while undergoing MEG. Handedness was assessed using the Edinburgh Handedness Inventory. *https://doi.org/10.1016/0028-3932(71)90067-4).

**Paradigm**. The experiment was conducted using Presentation (Neurobehavioral Systems) software. Figure 1a illustrates the experimental procedure. Before the start of the experiment, each participant completed a staircase procedure (2-up-1 down) varying the duration of the blank interval after the stimulus to maintain a detection rate of ~71% correct trials in the actual experiment. For the experiment, participants were instructed to focus their attention on the centre of the screen in order to discriminate the direction of an arrow (left or right). They completed several practice trials to familiarize themselves with the procedure. Prior to the target stimulus, a fixation cross with a uniformly variable duration (1500–1700 ms) was presented. Following this, the target (an arrow pointing either to the left or the right) was presented for 1 frame (corresponding to 16.7 ms [60 Hz refresh rate] for the patients and 8.3 ms [120 Hz refresh rate] for the healthy participant sample). After the arrow, a blank screen was presented. The duration of the blank screen was determined by the staircase procedure described above. At the lower end of the staircase (less than 1 frame), the blank screen was omitted. Following the blank screen, a mask consisting of an overlay of both arrows appeared for 500 ms. This mask ensures that the brain perceives the stimulus for the same amount of time across trials, as the presentation of said mask minimises retinal after-effects and post-stimulus visual processing[59]. Subsequently, a question mark prompted the participants to indicate the direction of the arrow by pressing one of two designated response buttons. The participants were instructed beforehand to always give a response, and in case of uncertainty, to guess. The participants were also instructed to respond as fast as possible. The response window lasted for 1500 ms, limiting the time window for each response. Every participant completed 6 blocks, each of which consisted of 72 trials. Participants were given the opportunity for a short break in between each block.

For patients, the mean hit rate across participants was 75.9% (s.d. 14.7%), and the mean reaction time was 872 ms (s.d. 203 ms). For the healthy controls, the mean hit rate across participants was 80.3% (s.d. 10.3%), and the mean reaction time was 750 ms (s.d. 78 ms).

**iEEG acquisition**. The two thalamic depth electrodes each had four intracranial electrode contacts (platinum–iridium contacts, 1.5 mm wide with 1.5 mm edge-to-edge distance). The clinically-relevant implantation target was the anterior thalamic nucleus. However, due its small size and the implantation trajectory, a subset of the electrode contacts invariably land in the mediodorsal thalamus (see Fig. 1b). All patients received bilateral implants, resulting in eight electrode contacts in the thalamic area. iEEG was recorded by feeding the signal into auxiliary channels of the MEG system, ensuring simultaneous recordings and synchronized triggers across iEEG and MEG. All recordings were continuously sampled at 678.17 Hz.

**iEEG electrode localisation**. We estimated the locations of these contacts using the Lead-DBS software[60]. First, we co-registered the post-operative CT scan to pre-operative T1-weighted image using a two-stage linear registration (rigid followed by affine) as implemented in Advanced Normalisation Tools[61]. Second, we spatially normalised these scans to MNI space based on the pre-operative T1-weighted image using the Unified Segmentation Approach as implemented in SPM12[62]. Third, we reconstructed the positions and trajectories of the DBS electrodes based on post-operative CT scan. Fourth, we corrected these reconstructions for brain-shift in post-operative acquisitions by applying a refined affine transform calculated between pre- and post-operative scans that were restricted to a subcortical area of interest (as implemented in the Lead-DBS software). Lastly, we visually confirmed the positions of the contacts using the DISTAL Atlas[63]. Full details of electrode positioning can be found in Supplementary Table 1. All analyses were performed separately on mediodorsal thalamic pairs, or anterior thalamic pairs.

**iEEG preprocessing**. The iEEG recordings underwent several steps to attenuate artifacts. All preprocessing steps were completed using the Fieldtrip toolbox[64]. First, we downsampled the iEEG recordings to 500 Hz. Second, we filtered the recordings using a 150 Hz Butterworth low-pass filter (order = 6), two Butterworth band-stop filters (to attenuate line noise; 49–51 Hz, 99–101 Hz; order = 6), and a 0.5 Hz Butterworth high-pass filter (order = 6). Third, we epoched the recordings around the onset of the visual target, starting 2 seconds before target onset and ending 2 seconds after target onset. Fourth, we inspected the recordings for

artifactual/epileptic activity, and any trials or channels exhibiting such activity were excluded (percentage of electrodes removed: 33.3% [+/− 21.1%]; percentage of trials removed: 15.6% [+/− 6.7%]).

**iEEG re-referencing**. Following artifact rejection, we re-referenced the iEEG recordings using a bipolar re-referencing montage to provide a measure of spatially-specific activity within the anterior and mediodorsal thalamic nuclei. All six patients had at least one bipolar-referenced electrode pair within the mediodorsal thalamus, and five of these patients had at least one bipolar-referenced electrode pair within the anterior thalamus. We first identified all bipolar pairs that would feasibly capture mediodorsal/anterior thalamic activity of a given participant, and then selected the pair which produced the cleanest mediodorsal/anterior thalamic evoked response (see Supplementary Fig. 12 for evoked response of the selected pairs). As we used post-stimulus evoked activity as our selection criteria, and our main analyses focused on the pre-stimulus window, we can assume that this selection procedure did not introduce issues of circularity into our main analyses[65]. Full details of bipolar electrode positioning and pairing can be found in Supplementary Table 1.

**Patient MEG acquisition and preprocessing**. We recorded MEG with a 248-channel whole-cortex magnetometer (MAGNES 3600, 4D Neuroimaging, San Diego, USA) in a magnetically shielded room. Patients sat upright in the MEG. All recordings were continuously sampled at 678.17 Hz. MEG data of patients 1, 2 and 3 were DC recorded, MEG data of patients 4, 5 and 6 was recorded with a bandwidth of 0.1–200 Hz. We digitised the patients' nasion, left and right ear canal, and head shape prior to each session with a Polhemus 3Space Fasttrack.

The recordings underwent several steps to attenuate artifacts. All preprocessing steps were completed using the Fieldtrip toolbox[64]. First, we downsampled the MEG recordings to 500 Hz. Second, we filtered the recordings using a 150 Hz Butterworth low-pass filter (order = 6), two Butterworth band-stop filters (to attenuate line noise; 49–51 Hz, 99–101 Hz; order = 6), and a 5 Hz Butterworth high-pass filter (order = 6). This high-pass filter was set at 5 Hz as slower-frequency activity (i.e., <5 Hz) was corrupted by movement-related artifacts introduced by the presence of iEEG recording equipment within the dewar (note: to address concerns that the phase bifurcation effect in the medial prefrontal cortex was artifactually driven by this filter, we also analysed an independent set of MEG data from healthy participants were a less aggressive filter was used [0.5 Hz; see below]). Third, we epoched the recordings around the onset of the visual target, starting 2 seconds before target onset and ending 2 seconds after target onset. Fourth, we denoised the MEG recordings by conducting PCA on reference channels (as implemented in the Fieldtrip function *ft_denoise_pca*). Fifth, we used ICA to detect and remove spatially-stationary artifacts including eye blinks, eye movements, cardiac artifacts, and residual motion related artifacts. Sixth, we inspected the recordings for artifactual/epileptic activity. Any trials/sensors exhibiting such activity were excluded (percentage of sensors removed: 38.6% [+/− 7.1%]; percentage of trials removed: 45.0% [+/− 10.8%]; see next paragraph for notes on these high percentages). Lastly, we reconstructed the preprocessed data in source space using individual head models and structural (T1-weighted) MRI scans. We reconstructed the time-locked MEG data using a single-shell forward model and a Linearly Constrained Minimum Variance beamformer (LCMV[66];), with the lambda regularisation parameter set to 5%.

It is important to note that the externalised wires of the intracranial electrodes introduced substantial noise into the MEG recordings, with many posterior MEG sensors becoming saturated as a result of noise. Across patients, few sensors remained over parietal and occipital regions (see Supplementary Fig. 7 for a topographic plot of artifactual sensors). We therefore refrain from drawing major conclusions based upon results observed in posterior sources.

**Healthy control MEG acquisition and preprocessing**. For the healthy control subjects, we recorded MEG with a 306-channel whole-cortex magnetometer (Elekta Neuromag TRIUX, Elekta, Stockholm, Sweden) in a magnetically shielded room. Participants sat upright in the MEG. All recordings were sampled at 2000 Hz and online-filtered with a pass-band of 0.1–660 Hz. Headshape was digitized analogue to patient's measurements.

As above, we downsampled the MEG recordings to 500 Hz. Second, we filtered the recordings using a 165 Hz Butterworth low-pass filter (order = 6), two Butterworth band-stop filters (to attenuate line noise; 49-51 Hz, 99–101 Hz; order = 6), and a 0.5 Hz Butterworth high-pass filter (order = 6). Third, we epoched the recordings around the onset of the visual target, starting 2 seconds before target onset and ending 2 seconds after target onset. Fourth, we used ICA to detect and remove spatially-stationary artifacts including eye blinks, eye movements, cardiac artifacts, and residual motion related artifacts. Fifth, we inspected the recordings for artifactual activity. Any trials/channels exhibiting such activity were excluded. LCMV beamforming was conducted in the same manner as described above.

**Phase bifurcation analyses**. All subsequent analyses were conducted using a combination of in-house custom code (available here: https://github.com/StaudiglLab/corticothalamic-connect) and the Fieldtrip toolbox. In instances where

we relied on custom code, the key equations are given. In instances where we used prebuilt Fieldtrip functions, those functions are explicitly named.

In the first instance, we asked whether the phase of pre-stimulus low-frequency band activity within the mediodorsal thalamus predicts visual detection. First, we estimated the phase of the pre-processed mediodorsal thalamic recordings using 6-cycle wavelets (33 linearly spaced estimates ranging from −800ms to 800 ms [that is, sampled every 50 ms]; for frequencies ranging from 5 to 20 Hz [in steps of 1 Hz]) Note that we expanded beyond the pre-stimulus window for the purpose of data visualisation (e.g., see Fig. 1c). Second, we split trials into two conditions based on whether the response on said trial was correct (from here on termed "hits") or incorrect (from here on termed "misses"). Third, we computed the phase bifurcation index (PBI) as described by Busch and colleagues (2009). Here, inter-trial phase clustering [ITPC; also termed 'phase locking value' ('PLV'); see Eq. (1)] for each condition was computed separately ($ITPC_{hits}$ and $ITPC_{misses}$), as well as inter-trial phase clustering for both conditions combined ($ITPC_{combined}$). The ITPC values were then used to estimate phase bifurcation [see Eq. (2)].

$$ITPC = \left| n^{-1} \sum_{r=1}^{n} e^{ik_r} \right| \qquad (1)$$

where: $n$ = number of trials, and $k$ = phase angle.

$$PBI = (ITPC_{hits} - ITPC_{combined}) * (ITPC_{misses} - ITPC_{combined}) \qquad (2)$$

It is worth noting that this measure suffers a trial number bias: conditions with fewer trials see higher scores than conditions with more trials. To address this, we created a shuffled baseline in which every trial was circularly shifted in time (preserving signal autocorrelation) by a random number of samples and the phase bifurcation index was recalculated using this shuffled data (1000 permutations). This shuffled baseline retained the trial imbalance present in the initial calculation, and retained the phase structure of every trial, but should no longer exhibit any phase clustering beyond what would be expected by chance. We then z-transformed the PBI derived from the real data using the mean and standard deviation of the permutations of the shuffled baseline to give an estimate of phase bifurcation relative to chance.

For statistical analysis, we pooled together the z-transformed PBI of each patient and conducted a group-level, cluster-based, permutation test[67] (using 64 permutations; i.e., every possible permutation from a sample of six patients [$2^6$]). To aid in the interpretability of the cluster (that is, one cannot state exact when a significant cluster arises, only that has arisen in the time-frequency window analysed; see ref. [68]), we restricted the cluster analysis to the pre-stimulus period (i.e., −800 ms to stimulus onset) and to the frequency range where this effect has been observed in previous studies of the cortex (i.e., 6–14 Hz; see ref. [69] for meta-analysis). Cluster analysis addressed issues of multiple comparisons across time and frequency while the spectrotemporal region of interest ensured spectral/temporal specificity to pre-stimulus low-frequencies. As we only used a single mediodorsal thalamic channel (derived from a bipolar-referenced electrode pair) from each participant for this analysis, there were no multiple comparisons across space.

To supplement the main statistical result, we report the Bayes Factor at the peak voxel. Bayes factor was computed using the *bayesFactor* toolbox (https://github.com/klabhub/bayesFactor). We selected a default prior for the Bayesian t-test (i.e., the Cauchy prior [$2/\sqrt{2}$])[70].

To address the issue of the wavelet-induced smearing of a post-stimulus effect into the pre-stimulus window, we repeated the statistical analysis as above, but with the exclusion of any pre-stimulus sample point where the edges of the wavelet (for a given frequency) would extend into the post-stimulus window. After excluding the pre-stimulus time bins that could be compromised by wavelet-induced temporal smearing of a post-stimulus effect, phase bifurcation continued to be observed (mean cluster $t(5) = 2.65$, $p_{clus} = 0.047$, $BF_{10} = 22.25$).

We repeated the entirety of this analytical pipeline for the anterior thalamic recordings.

We then applied this same approach to the source-reconstructed MEG data. As before, the z-transformed phase bifurcation index for each participant was pooled and subjected to a group-level, cluster-based, permutation test (this time using the Fieldtrip function *ft_sourcestatistics*). When statistically appraising phase bifurcation in the patient MEG data ($n = 6$), 64 permutations were used once again. As the function *ft_sourcestatistics* cannot conduct cluster analyses across time/frequency while simultaneously conducting analyses across space, we averaged the PBI values across the pre-stimulus window (i.e., −800 ms to stimulus onset) and across the frequency range where this effect has been observed in previous studies of the cortex (i.e., 6–14 Hz; see[69] for meta-analysis), which provided a single PBI value for each voxel of source-reconstructed MEG data. The cluster analysis was then conducted across space on this time/frequency averaged data. We repeated the process for the healthy control MEG ($n = 12$), however, 4096 permutations were used (i.e., $2^{12}$ permutations) in place of 64 permutations.

**Phase reset analysis**. To test whether the phase of ongoing activity resets following stimulus onset, we computed low-frequency spectral power (6 to 9 Hz; in steps of 1 Hz) across the epoch (−800 ms to 800; in steps of 25 ms) using 6-cycle wavelets, and then took the average 'pre-stimulus' power just before stimulus onset (−200 to 0 ms) and 'post-stimulus' power just after stimulus onset. We conducted this spectral decomposition twice: first, on single trials before averaging the result

across trials (i.e., total power), and second, on the trial-averaged amplitude (i.e., evoked power). If phase does reset after stimulus onset, then phase should align across trials after stimulus onset, and will present as an increase in evoked power for post-stimulus activity relative to pre-stimulus activity. In contrast, no change in total power will be observed on the single trial level. To statistically appraise the effect, we conducted a 2 × 2 repeated measures ANOVA to probe how spectral power changed as a function of epoch (pre- vs. post-stimulus) and decomposition method (single trial decomposition vs. trial-averaged decomposition).

Note that while phase clustering metrics are also sensitive to phase resets, they are not specific (that is, a spike in phase clustering after stimulus onset may reflect a phase reset, but may also reflect an evoked response). In contrast, the approach used here can be both sensitive and specific to phase resets, as the evoked response component would be consistent across the total- and evoked power metrics.

**Inter-site phase clustering connectivity analyses**. To assess whether the mediodorsal thalamus couples with the cortex prior to visual perception, we examined inter-site phase clustering (ISPC) between the thalamic recordings and the source-reconstructed MEG recordings. First, we estimated oscillatory phase using wavelets (no parameters were changed from the phase bifurcation analyses described above). Second, we computed the circular distance between the instantaneous phase angle in the thalamus and the phase angle in the source-reconstructed voxel (individually for every trial, timepoint, frequency and source-reconstructed voxel). We then computed ISPC clustering over trials [see Eq. (3); note that this is identical to Eq. (1), with the exception that it uses the phase angle difference between two regions, rather than a single, observed phase angle].

$$ISPC = \left| n^{-1} \sum_{r=1}^{n} e^{id_r} \right| \qquad (3)$$

where: $n$ = number of trials, and $d$ = circular distance between phase angles.

To examine whether the observed ISPC differed from chance, we generated a distribution of chance ISPC values by randomly shuffling the trials of the thalamus recordings relative to the MEG recordings and re-computing the ISPC (total permutations = 1000). We then z-transformed the observed ISPC using the mean and standard deviation of the chance distribution (as done for the PBI measure). Statistical analysis matched that of the PBI analyses on the source-reconstructed MEG signal (that is: cluster-based permutation tests with a specific focus on pre-stimulus low-frequency activity).

To evaluate whether this connectivity varied as a function of perceptual performance, we calculated ISPC for hits and misses separately, with a subsampling procedure used for hits to ensure trial numbers were balanced across the two conditions. We then directly contrasted the resulting ISPCs in a cluster-based permutation test (again, across voxels using *ft_sourcestatistics*, with each voxel matching the value of the average of low-frequency [6–14 Hz], pre-stimulus [−800 to 0 ms] ISPC for that voxel).

It is worth noting that the ISPC can be biased by volume conduction. In such instances, the phase lag between the thalamus and source-reconstructed MEG should cluster heavily around 0 or 180 degrees. This was not the case in our data (see Fig. 2c). Residual concerns about spurious corticothalamic coupling are addressed by our "phase slope index" analysis below, which excludes zero-lag angle differences from the computation.

**Phase slope index analyses**. To assess the directionality of the coupling between the mediodorsal thalamus and cortex, we used the phase slope index[29]. To this end, we calculated the Fourier spectrum of the pre-stimulus signal (−800 to 0 ms) using a Hanning tapered FFT approach, and used the resulting signal to compute the PSI (as implemented by the function *ft_connectivity_psi* in the Fieldtrip toolbox). As before, we compared the observed PSI to chance by shuffling the trials of the thalamus recordings relative to the MEG recordings and re-computing the PSI (total permutations = 200 [the number of permutations were reduced relative to the analyses above due to computational limitations]). We then z-transformed the observed PSI using the mean and standard deviation of the chance distribution (as done for the PBI and ISPC measures). Statistical analysis matched that of the PBI and ISPC analyses on the source-reconstructed MEG signal.

We repeated this approach for hits and misses separately. The resulting z-transformed PSI measures were directly compared in a group-level, cluster-based, permutation test.

**Mediation analyses**. To assess the possible mediating effect of the mediodorsal thalamus, we first set out to measure phase bifurcation on the single-trial level. As the phase bifurcation index relies on data from all trials, such an approach cannot be used to create trial-level models of mediation. Instead, for a given patient, and for every pre-stimulus sample point, we took the mean phase angle across all "hit" trials, and then derived the mean resultant vector between this hit-averaged phase angle and the observed angle on a given trial ("hits" and "misses"). This provides a value between 0 and 1 which indicates how close the given trial was to the optimal phase for subsequent visual detection [the higher the value, the closer the phase][23].

We then used a series of patient-specific regression models to assess (1) whether the distance to the optimal phase within the medial prefrontal cortex predicts visual detection (independently of the mediodorsal thalamus) [see Eq. 4], (2) whether the

distance to the optimal phase within the medial prefrontal cortex predicts the distance to the optimal phase within the mediodorsal thalamus [see Eq. 5], and (3) whether the distance to the optimal phase within both the mediodorsal thalamus and the medial prefrontal cortex, in combination, predicts visual detection [see Eq. 6].

$$Y = j_1 + cX \qquad (4)$$

$$M = j_2 + aX \qquad (5)$$

$$Y = j_3 + c'X + bM \qquad (6)$$

Where Y represents perceptual outcome (either hit or miss), X represents distance to optimal phase in the medial prefrontal cortex, M represents distance to optimal phase in the mediodorsal thalamus, and j represents the intercepts. When predicting Y, logistic models were used. When predicting M, linear models were used. As the scaling of coefficents differs between these two models, all coefficents were standardised by dividing by the standard error of fit. This brought both forms of coefficents into the same unit space.

While, in theory, one can test this at every time, frequency and source-reconstructed voxel, this is prohibitively computationally expensive (~14 days on our hardware). In addition, it is debatable as to whether any meaningful measure of mediation can be derived from moments (be that timepoints, frequencies or voxels) where the independent or mediator variable does not reliably predict the dependent variable[71]. Therefore, for the sake of computational efficiency and statistical validity, we restricted our analyses to the moments in which phase bifurcation peaked in the medial prefrontal cortex and mediodorsal thalamus. While such an approach would inflate the likelihood of finding a link between physiology and behaviour, given that the purpose of this analysis is to compare the relative link of the mediodorsal thalamus and prefrontal cortex to behaviour (as opposed to the absolute link to behaviour), we do not believe that this is a concern.

In our first test for mediation, we assessed whether the indirect effect (i.e., the *ab* pathway in Fig. 3a) differed significantly from zero. The indirect pathway describes the extent to which mediodorsal thalamic phase bifurcation explains the impact of medial prefrontal cortical phase bifurcation on perceptual performance. Thus, if this is significantly greater than zero, one can conclude that the influence of the medial prefrontal cortex on perceptual performance is mediated by the mediodorsal thalamus in some way, shape or form. To this end, we operationalised the indirect effect as the product of t-statistics of a and b, normalised by the variance (see Eq. 7, taken from ref.[72]).

$$ab = \frac{t_a t_b}{\sqrt{t_a^2 + t_b^2 + 1}} \qquad (7)$$

Where $t_a$ and $t_b$ are the standardised coefficients derived from Eq. 5 and Eq. 6 respectively. We then *z*-transformed the magnitude of this effect using the mean and standard deviation of chance-level indirect effects (which were calculated by shuffling the trials of the mediodorsal thalamic recordings relative to the behavioural and medial prefrontal measurements and recomputing the regression models; 1000 permutations). We then pooled the *z*-transformed measure of the indirect effect of each patient and contrasted them against the null hypothesis that the indirect effect was no greater than chance (i.e., $z = 0$) in a permutation-based *t*-test. Here, for each permutation, the sign of each patient's *z*-transformed indirect effect was randomly assigned, and the *t*-values were recomputed. The *p*-value was then derived by comparing the "true" *t*-value to this surrogate distribution.

In our second test of mediation, we asked whether the influence of medial prefrontal activity on perceptual performance is diminished after accounting for mediodorsal thalamic activity. To this end, we contrasted the "total effect" (*c* in Eq. 4) against the "direct effect" (*c'* in Eq. 6). If the direct effect is significantly smaller than the total effect, one can infer that the second regressor in Eq. 6 (i.e., the distance to the optimal phase in the mediodorsal thalamus) has a mediating influence over prefrontal contributions to visual detection. As above, we *z*-transformed the observed difference between the total and direct effects using the mean and standard deviation of chance-level differences (which were calculated by shuffling the trials of the medial prefrontal and mediodorsal thalamic recordings relative to the behavioural data and recomputing the regression models; 1000 permutations). We then pooled the *z*-transformed difference of each patient and contrasted them against the null hypothesis that there was no difference between the total and direct effects (i.e., $z = 0$) in a permutation-based *t*-test.

We supplemented the mediation analysis with an approach based on partial correlations (see Supplementary Fig. 11). We computed the single trial measures of distance to the optimal phase as above, but rather than using logistic models to assess the relationship between brain activity and perceptual performance, we used correlations and partial correlations. Specifically, we computed a Spearman's Rank correlation between the distance to the optimal medial prefrontal low-frequency phase and perceptual performance, and a partial Spearman's Rank correlation between the distance to the optimal medial prefrontal low-frequency phase and perceptual performance while accounting for the distance to the optimal mediodorsal thalamic low-frequency phase.

Note that, while mathematically plausible, inverting the mediation model such that the mediodorsal thalamus becomes the independent variable and the medial prefrontal cortex becomes the purported mediator would be conceptually invalid as our PSI analyses have demonstrated that the cortex precedes the thalamus, and mediation analyses rest upon the assumption that the mediator follows the independent variable in time[73]. In other words, event A cannot mediate the influence of event B on event C if neither event B nor C have happened yet.

**Reporting summary**. Further information on research design is available in the Nature Research Reporting Summary linked to this article.

## Data availability
Source data are provided with this paper, at are available at osf.io/n9w36/. The raw data are protected and are not openly available due to data privacy laws, though (subject to these privacy laws) the data are available upon reasonable request.

## Code availability
The code used for this study is available at https://github.com/StaudiglLab/corticothalamic-connect[74] (https://doi.org/10.5281/zenodo.6457779).

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

## Acknowledgements

We are indebted to all patients who volunteered their time to participate in this study. T.S. is funded by the European Research Council, Starting Grant 802681. S.H. is funded by the European Research Council, Consolidator Grant 647954, and the Economic Social Sciences Research Council (ES/R010072/1).

## Author contributions

Conceptualization: T.Z., S.H., T.S.; Resources: T.Z., F.C.S., J.V.; Investigation: T.Z., S.R., T.S.; Formal Analysis: B.J.G., T.S.; Funding acquisition: S.H., T.S.; Project administration: T.Z.; Writing—original draft: B.J.G., T.S.; Writing—review and editing: B.J.G., T.Z., S.R., F.C.S., J.V., S.H., T.S.

## Funding

## Competing interests

The authors declare no competing interests.
