## [Peer Review File · Nature Communications]

Rhythmic interactions between the mediodorsal thalamus and prefrontal cortex precede human visual perceptionREVIEWER COMMENTS

Reviewer #1 (Remarks to the Author):

This study examines predictors of perception in brain oscillations in a rare data set combining MEG recordings and simultaneous intracranial EEG from the mediodorsal nuclei of the thalamus in 6 patients while they are undergoing a visual detection task, complemented by MEG recordings in 12 healthy participants in the same task. The results reveal phase (roughly in the alpha-band) to predict detection performance, both in the mediodorsal thalamic nuclei and prefrontal cortex. Alpha power was not predictive. Connectivity between these sites (phase locking) was also significant in the alpha band. A mediation analysis identified significant contribution of thalamic activity to perceptual performance via prefrontal activity. No link between thalamic – prefrontal connectivity and performance were observed. The main message is that thalamocortical interactions need to be taken into account if we were to resolve how prefrontal activity contributes to cognition.

This is an interesting study on pre-stimulus predictors of perception with a new perspective because taking into account thalamic activity. The manuscript is well written. The contributions are novel and an interesting hypothesis is presented how the orchestration of alpha phase across thalamo-prefrontal sites may contribute to perception and explain its phase dependency. I only have a few points, none fundamental.

Specific points

1) There is a discrepancy in the frequency at which pre-stimulus phase bifurcation predicts perception, with thalamic frequency being below 8 Hz (Fig 1c), and prefrontal frequency being above 10Hz both in the patients (Fig 1f) and healthy participants (Fig 1g). This is surprising as one would expect a good frequency match if these signals were to represent activity from the same network. It would seem important that this mismatch can be explained and/or discussed.

2) It is puzzling that the phase opposition analysis in the MEG data did not reveal any areas in the dorsal attention network (FEF, parietal cortex) to predict perception, as this is the most frequent observation in the literature, right? Or has this simply not been reported for this data set? Could this be addressed by showing all results across the brain, and if no areas are detected outside the prefrontal cortex, by discussing this discrepancy with the literature.

3) A suggestion is made how thalamo-cortical interactions could contribute to perception mechanistically (some sort of prediction model, lines 133-144). One wonders how this relates to the alpha-predictive coding model of Alamia and VanRullen (see below). Could this be addressed?

Alamia A, VanRullen R. Alpha oscillations and traveling waves: Signatures of predictive coding? PLoS Biol. 2019 Oct 3;17(10):e3000487.

Reviewer #2 (Remarks to the Author):

My understanding: 6 patients performed a visual detection task and the authors recorded thalamic iEEG and whole head MEG simultaneously. The authors are claiming that MD thalamus mediates the dependence mPFCs contribution to the visual detection, and are indexing this mediation using the phase bifurcation index used in Busch Dubois VanRullen J Neuro 2009. I agree that it is interesting data, but I am unsure how exactly this adds to our overall understanding in it's current form. I have a few concerns about its contribution I would like the authors to address to help better understand the data

point 1:

I think the question of "does thalamus impact human cognition" is actually a rather well explored question in the context of visual perception (such as in the various S Kastner Lab papers cited by the authors). Are MD <-> PFC circuits relationship is related to pulvinar, LGN and primary visual cortex alpha rhythms. Alpha in these regions, which are theoretically upstream of mPFC, have also been implicated in visual perception. The schematic in Fig 3 left me wondering about this.

As a control for this, perhaps the authors could perform a similar PSI analysis using a searchlight for some of your key metrics across your sensors with >4 or 5 participants, to ensure that we understand the brain network involved? Or, at least, sensors from the P03/P07 and CPz EEG coordinate regions, where from what I can tell there are several participants in supp fig 7

point 2:

I would have liked to see Fig 1C, Fig 2E, and Supp fig 4 and 5 to have included some more regions to develop an account of the network involved in this alpha-rhythmic influence of perception. could the authors create 2 topographic maps: 1 that shows the PBI distribution using Frontal cortex as a seed, and another using MD thalamus as a seed? (e.g. more red regions show higher PBI)

point 3:

The sentence "Notably, the ideas presented above do not explain the rhythmicity of our observed effect." on page 6 is confusing. If alpha band connectivity is not an important signal in these regions and for this task, why define your metrics using it? There are other signals in EEG to corroborate this claim. This is especially worrying since most of the stronger effects (Fig 1C, 2A, Supp Fig 4, Supp fig 2) are sometimes closer to what most would call theta band. So, again, call it alpha if this is not important?

One specific thought: The data all hinge on alpha band waveform characteristics, I think, which is slightly confusing to me. What about using raw or bandpass filtered neural signals across a 4-20 Hz range in thalamus and mPFC to classify whether a target was reported as detected or not? This could be done by on a time point by timepoint basis using logistic regression for example, as long as the authors make sure to resample trials to equalize incorrect and correct ones in the training set.

point 4:

"No robust phase bifurcation was observed in additional anterior thalamic recordings ($t(4) = 4.20$, $p_{\text{clus}} > 0.5$, $\text{BF}_{10} = 5.63$; though no difference in PBI was observed between the anterior and mediodorsal thalami: $t(5) = 7.52$, $p_{\text{clus}} = 0.094$, $\text{BF}_{10} = 25.84$; see supplementary figure 2)." pg 3. This is a bit troubling, and note that the bottom panel of supplementary figure two is really showing a difference in the theta range. Can the authors offer more information as to why this is?

When I went to the Methods to better understand this, I read "For statistical analysis, the z-transformed PBI of each patient were pooled together and subjected to a group-level, cluster-based, permutation test (using 64 permutations; i.e., every possible permutation from a sample of six patients [26])." Could the authors better explain the extent of the clusters and the dimensionality of their thalamic and MEG recordings? It was unclear whether this cluster correction was occurring over spatial regions of the brain or over frequencies?

I also found it slightly difficult to find out how many iEEG contacts where in the thalamus? Could the authors clarify. More generally, The methods are slightly difficult to read at times.

smaller things

1)

Is the order of the Butterworth filter somewhere? I missed it if so. All I see on page 8 "Second, the recordings were filtered using a 150Hz Butterworth low-pass filter, two Butterworth band-stop filters (to attenuate line noise; 49-51Hz, 99-101Hz), and a 0.5Hz Butterworth high-pass filter"

2)

Fig 1(e) "Patient-specific observed phase-bifurcation (black line) compared to a surrogate distribution (histogram) for individual peak bifurcation frequencies"

- P2 seems to have a rather large PBI and low frequency compared to everyone else. Could the authors create the equivalent fig 1C without that participant?

- Additionally, could the authors label Fig 1(d) with the subplots labeled by participant?

3)

Page 6: "was diminished after accounting for pre120 stimulus thalamic activity ($t(5) = 2.26$, $p = 0.031$, $BF_{10} = 3.06$)" This claim cannot be made using that statistical test. Can the authors directly compare the conditions instead?

4)

Fig 3B is never referenced?

5)

I would like more discussion of the masking paradigm, why it was employed, and how it relates to point 2

6)

The authors state that The medial prefrontal cortex leads the mediodorsal thalamus uniquely for hits in Figure 2. Can the fig 3 schematic somehow incorporate this?

7)

Can the authors characterize how alpha power changes in the MD thalamus and PFC?

Reviewer #3 (Remarks to the Author):

This is a very interesting study that links thalamic activity in humans (the measurement of which is rare and novel), and its connections with the prefrontal cortex with perception. Using a simple discrimination of a briefly presented then masked visual stimulus, the authors show that, depending on whether the stimulus was correctly identified, the phase of the alpha-band (6–9 Hz) oscillations in the mediodorsal thalamus, as measured in patients with intracranial electrodes implanted, differed approximately 300–800 ms before the onset of the stimulus. The authors describe this "phase bifurcation" also in a broader frequency range in MEG in the prefrontal cortex in both the patients and a control group.

The surprising result is that whether the subject correctly discriminated the visual stimulus did not depend on the alpha phase at the time the stimulus was presented, but rather it depended on the oscillation phase at a substantial time *before* the stimulus was presented.

I have a few specific comments:

1. The manuscript is a bit confusing on this point. It seems based on Figure 1d that the oscillation phase was always consistent at time zero, when the stimulus was presented. That is, the oscillations were always in phase with each other whether or not the stimulus was detected. One might expect that the phase would just be random here, but Figure 1d shows only a single phase. Shouldn't the amplitude drop to zero here if the phases were random? Or is the amplitude normalization performed at each time point somehow? It might be helpful to show a distribution of phases at each time point, for each condition. It's not entirely clear, for example, that a particular phase is required some 400 ms before the stimulus, or that the two conditions just tended to differ in phase during this epoch.

2. The manuscript states that the phase bifurcation index (PBI) was "peaking at 8 Hz, 300 ms to stimulus onset). However, in Figure 1c, there seems to be a larger, more significant, cluster of positive PBI at about 7 Hz, 500 ms before stimulus onset.

3. In some subjects, there seems to be a rapid (less than 100 ms) and noticeable change in phase, immediately after the stimulus onset. Is this consistent across subjects?

4. There seems to be a sizable difference in the PBI in the MEG between the patient and control groups, after the stimulus onset. Specifically the controls show consistently negative PBI (it is not clear how to interpret negative PBI). The positive PBIs also differ somewhat in their timing pre-stimulus (about 400 ms for the patients and 300 ms in controls). Is this significant?

5. In Figure 2c, the caption says that the patient's orange lines should be drawn at different lengths, with the length indicating that patient's consistency. But all of the orange lines have the same length.

6. The peak PBI is around 7 Hz in the thalamus (iEEG) but something like 11 Hz in the prefrontal cortex (MEG). How are the phase-lags and connectivity measures computed between the two methods if the peak frequencies are not the same? I don't see how it makes sense. If it's not the same frequency, are you sure you are looking at the same phenomenon?

Reviewer #4 (Remarks to the Author):

The manuscript by Griffiths et al. investigates the thalamocortical loop while participants perform a visual detection task.

The manuscript is well written and the methods of analysis seem state of the art.

Major comments:

The authors should elaborate on their re-referencing.

Were contacts in the thalamus re-referenced against a contact in white matter?

Were adjacent contacts chosen to form bipolar pair?

How were pairs chosen for further analysis? Only one pair per participant? Only mediodorsal thalamic nuclei? Were pairs in anterior thalamic nuclei equally accepted? Should this be mentioned in the discussion?

As the authors deem direct recordings of the thalamus and cortex “incredibly rare”, they might want to cite Sarnthein&Jeanmonod 2007 and 2008.

Minor comment

Six patients (four female, ...

Point-by-Point Reply: NCOMMS-21-32345A

We thank the four reviewers for their thoughtful and constructive comments. We address each of the reviewer's point individually below. For your convenience, changes to the manuscript are highlighted in green.

Reply formatting guide:

Reviewer remarks: **bold, black.**

Author response: *normal, blue.*

Manuscript quotation: *italics, blue.*

Reviewer #1 (Remarks to the Author):

This study examines predictors of perception in brain oscillations in a rare data set combining MEG recordings and simultaneous intracranial EEG from the mediodorsal nuclei of the thalamus in 6 patients while they are undergoing a visual detection task, complemented by MEG recordings in 12 healthy participants in the same task. The results reveal phase (roughly in the alpha-band) to predict detection performance, both in the mediodorsal thalamic nuclei and prefrontal cortex. Alpha power was not predictive. Connectivity between these sites (phase locking) was also significant in the alpha band. A mediation analysis identified significant contribution of thalamic activity to perceptual performance via prefrontal activity. No link between thalamic – prefrontal connectivity and performance were observed. The main message is that thalamocortical interactions need to be taken into account if we were to resolve how prefrontal activity contributes to cognition.

This is an interesting study on pre-stimulus predictors of perception with a new perspective because taking into account thalamic activity. The manuscript is well written. The contributions are novel and an interesting hypothesis is presented how the orchestration of alpha phase across thalamo-prefrontal sites may contribute to perception and explain its phase dependency. I only have a few points, none fundamental.

We would like to thank the reviewer for taking the time to review the manuscript, and for the positive appraisal. We hope our responses clarify the issues raised.

Specific points

1) There is a discrepancy in the frequency at which pre-stimulus phase bifurcation predicts perception, with thalamic frequency being below 8 Hz (Fig 1c), and prefrontal frequency being above 10Hz both in the patients (Fig 1f) and healthy participants (Fig 1g). This is surprising as one would expect a good frequency match if these signals were to represent activity from the same network. It would seem important that this mismatch can be explained and/or discussed.

This is indeed an interesting question about the data (and one also posed by Reviewer #3). Speculatively, this may relate to the concept of travelling waves (Muller et al., 2018): travelling waves generated by weakly coupled oscillators tend to propagate from regions with faster oscillating frequencies to regions to slower intrinsic frequencies (Ermentrout

& Kopell, 1984). In the context of the data presented here, the shift in frequency between the prefrontal cortex and mediodorsal thalamus perhaps provides the necessary conditions to propagate local prefrontal signals as a travelling wave to the mediodorsal thalamus (which aligns with our observation of directed connectivity from the prefrontal cortex to the thalamus). In other words, physiologically speaking, it may be better for the two regions to have somewhat asynchronous frequencies to allow for directional connections, as opposed to having perfectly aligned frequency bands. We have elaborated upon this in the discussion (see page 8, lines 207-227).

“One may be wondering why prefrontal cortical and mediodorsal thalamic phase bifurcation arose at neighbouring, rather than identical, frequencies (~11Hz and ~8Hz respectively). While the spectral smearing incurred through the use of wavelets for our measure of inter-site phase clustering and the 6Hz bandwidth used for the phase-slope index analyses provide a mathematical explanation of connectivity between the two differing frequency bands, it wouldn’t explain the physiological underpinnings of such a phenomenon. We speculate, however, that the observed connectivity in conjunction with the mild difference in frequency may relate to travelling waves (e.g. ^{40,41}); more specifically, travelling waves that come about through weakly-coupled oscillators⁴². Models of weakly-coupled oscillators suggest that travelling waves can couple two regions so long as the oscillator of the transmitting region has a higher intrinsic frequency than the oscillator of the receiving region. In the case of the data presented here, we would anticipate that a travelling wave would begin within the prefrontal cortex (given its higher peak phase bifurcation frequency) and propagate to the mediodorsal thalamus. Notably, such an idea neatly ties to the phase-slope index results which demonstrated directed connectivity from the prefrontal cortex to the mediodorsal thalamus. Moreover, this explanation also aligns with the “Bayesian observer” described above, and the travelling waves inherent in such a hypothesis³⁶. Of course, this remains a speculative interpretation of the frequency differences between the two regions as very little is known about corticothalamic travelling waves. Consequently, such an explanation presents a novel avenue for future research regarding corticothalamic interactions, and may provide an answer as to why two regions with differing bifurcating frequencies may relate to a shared phenomenon.”

2) It is puzzling that the phase opposition analysis in the MEG data did not reveal any areas in the dorsal attention network (FEF, parietal cortex) to predict perception, as this is the most frequent observation in the literature, right? Or has this simply not been reported for this data set? Could this be addressed by showing all results across the brain, and if no areas are detected outside the prefrontal cortex, by discussing this discrepancy with the literature.

We had avoided interpreting results from the posterior half of the patient MEG as the intracranial wires caused profound interference with those sensors (see supplementary figure 7 for depiction) and we were uncertain of the reliability of effects arising from regions such as the parietal cortex. Indeed, when focusing in on the dorsal attention network of the patient MEG, no substantial bifurcation can be observed. That said, when doing the same in the healthy control data, a local peak of phase bifurcation can be seen in the left frontal eye field, suggesting that the lack of an effect in the patient MEG may be driven by issues relating to the signal-to noise ratio of some sensors as opposed to an absence of an effect. We have added a supplementary figure to visualise this effect (see supplementary figure 8; also see supplementary figure 10 for similar source plots for the connectivity metrics), pointed to it in the results (see first quote below; see page 4, lines 100-104), and added an additional paragraph in the discussion to describe and evaluate these effects (see second quote below; see page 9, lines 253-263).

“Previous studies have also observed phase bifurcation over the dorsal attention network (e.g. ^{12,26}). While the positioning of the electrode wires during the patient MEG recording prevents us from reliably probing these more posterior sources (see supplementary figure 7), the healthy control MEG recordings show analogous results to those which have been reported previously (see supplementary figure 8).”

“Beyond the prefrontal cortex, numerous other cortical regions have been shown to engage in visual perceptual processes (e.g., the dorsal attention network; ^{12,26}). Due to the positioning of the iEEG wires in the MEG, however, we were unable to reliably record signals from these regions, and hence investigate how they interact with the mediadorsal thalamus. Despite this however, we observed interesting connectivity dynamics where the low-frequency thalamic activity seemingly leading low-frequency activity in the occipital cortex (see supplementary figure 10). In the context of the prefrontal connectivity patterns, one could speculate that signals from the prefrontal cortex pass to the occipital lobe via the mediadorsal thalamus, and may explain why phase opposition effects can be seen across the cortex ^{e.g.11,12,26}. Of course, given that these results depend on signals generated from sources with poor MEG sensor coverage, one must take these findings with a grain of salt.”

Supplementary Fig. 8. Previous work has demonstrated that the dorsal attention network shows similar phase bifurcation effects as the prefrontal cortex. Here, we explored whether similar effects can be observed in our data, taking two regions that have previously been shown to demonstrate rhythmic sampling: the frontal eye fields (panel a) and the intraparietal sulcus (panel b) [Fiebelkorn, Minsk & Kastner, 2019]. Panel A has been centred on MNI co-ordinates representative of the left and right frontal eye fields (Murd et al., 2020). Panel B has been centred on MNI co-ordinates representative of the intraparietal sulcus (Bray et al., 2013). While we did not observe any substantial phase bifurcation in either area for the patient data, substantial phase bifurcation was observed in the left frontal eye fields for the healthy controls. The absence of an effect in the patient data may be attributable to the fact that the externalised intracranial wires ran over the back of the head, producing large artifacts that reduce the signal-to-noise of data sampled from these regions. Consequently, the absence of an effect in the dorsal attention network of the patients should not be interpreted as an absence of an effect in the population, but simply an idiosyncratic feature of MEG recordings from patients with externalised DBS wires.

3) A suggestion is made how thalamo-cortical interactions could contribute to perception mechanistically (some sort of prediction model, lines 133-144). One wonders how this relates to the alpha-predictive coding model of Alamia and VanRullen (see below). Could this be addressed?

Alamia A, VanRullen R. Alpha oscillations and traveling waves: Signatures of predictive coding? PLoS Biol. 2019 Oct 3;17(10):e3000487.

Thank you for bringing this paper to our attention. It seems to neatly tie into the “Bayesian observer” interpretation we present in the discussion, as well as provide an explanation as to why such a Bayesian observer would be observed in metrics assessing corticothalamic alpha-band connectivity (an explanation we were previously lacking). It also aligns with our response to the reviewer’s initial point, where we suggest that the prefrontal cortex connects to the mediadorsal thalamus via a travelling wave. We have expanded the original paragraph discussing the thalamus as a Bayesian observer to include this work, and also highlight how the computational model may

explain the performance-dependent nature of the directional connectivity results (see page 7, lines 172-191).

“Notably, computational models suggest that these mechanistic interactions produce patterns of low-frequency travelling waves between the interacting regions³⁶, which may explain why corticothalamic connectivity was most prevalent in the low frequencies. If template updating were to breakdown, one could expect that the detection of a transient change in sensory input would fail and corticothalamic low-frequency connectivity would dissipate, which may explain why the directional connectivity from the prefrontal cortex to the mediodorsal thalamus observed here was performance-dependent. While the correlative nature of our data prevents us testing these ideas, future studies which disrupt corticothalamic interactions (e.g., through direct thalamic stimulation) could directly test the causal nature of these hypotheses.”

Reviewer #2 (Remarks to the Author):

My understanding: 6 patients performed a visual detection task and the authors recorded thalamic iEEG and whole head MEG simultaneously. The authors are claiming that MD thalamus mediates the dependence mPFCs contribution to the visual detection, and are indexing this mediation using the phase bifurcation index used in Busch Dubois VanRullen J Neuro 2009. I agree that it is interesting data, but I am unsure how exactly this adds to our overall understanding in it's current form. I have a few concerns about its contribution I would like the authors to address to help better understand the data.

We'd like to thank the reviewer for taking the time to provide insightful feedback about our manuscript. We hope our responses clarify the contribution of these results to the field. Specifically, we would suggest that these results add to the field by (1) demonstrating that electrophysiological thalamocortical interactions in humans contribute to visual perception, as previous work has focused on non-human primates, which are suboptimal models for human prefrontal contributions to cognition; (2) identifying a second thalamocortical network (i.e., prefrontal cortex to mediodorsal thalamus) that contributes to visual perception, challenging the existing mantra that it is posterior sources that drive such processes; and (3) identifying real-time, oscillatory thalamocortical interactions in humans, as many previous studies have relied on fMRI which lacks sufficient temporal precision to achieve either of these feats. We hope our responses below and the changes made to the manuscript help clarify these points.

point 1: I think the question of "does thalamus impact human cognition" is actually a rather well explored question in the context of visual perception (such as in the various S Kastner Lab papers cited by the authors). Are MD <-> PFC circuits relationship is related to pulvinar, LGN and primary visual cortex alpha rhythms. Alpha in these regions, which are theoretically upstream of mPFC, have also been implicated in visual perception. The schematic in Fig 3 left me wondering about this.

We would argue that the Kastner papers we have cited (i.e., Fiebelkorn et al., 2019; Saalman et al., 2012) are not a great example of research probing the question "does thalamus impact *human* cognition", because these papers have studied macaques. We don't mean to suggest that these findings are in no way applicable to humans, but would suggest that it is a logical leap to suggest that findings from the macaque pulvinar generalise to the human mediodorsal thalamus given that (1) these two thalamic nuclei have distinct patterns of anatomical and functional connectivity to the cortex, so functionality cannot be assumed to generalise, and (2) the principal connection of the MD, the PFC, is the brain region that has undergone the most transformation during the course of evolution (the human PFC accounts for 28.5% of the neocortex, whereas the macaque PFC accounts for only 11.3% of the neocortex; Passingham, 2009), so models examining thalamic interactions with the PFC derived from our evolutionary ancestors are much less likely to generalise to humans than previous work investigating the interactions between the pulvinar and parietal cortex. We have clarified these distinctions within the discussion (see quote below; see pages 8-9, lines 228-245).

"Our observation of low-frequency connectivity between the mediodorsal thalamus and prefrontal cortex suggests that humans exhibit similar thalamocortical loops to those observed in animals^{18,38}. To date, studies of these loops in humans are scarce⁴³, owing to the fact that simultaneous, direct recordings of the specific thalamic nuclei and cortex are rare (see⁴⁴⁻⁴⁷ for other examples recording from different thalamic nuclei). As such, to understand these moment-by-moment dynamics, the field has had to rely on generalising earlier findings from animal models to humans, rather than studying humans directly. While these models have provided fantastic advances into our understanding of the role of the

thalamocortical loops in visual perception, they do have their limitations. Firstly, many of these studies have focused on the pulvinar (e.g., ^{37,38}), whose anatomical and functional connections to the cortex are notably different to the cortical connections of the mediodorsal thalamus, meaning these results cannot be generalised to explain the role of the mediodorsal thalamus in visual perception. Second, animal models of the prefrontal cortex are limited in their generalisability relative to animal models of other cortical regions owing to the unique evolutionary divergence in structure of the prefrontal cortex ⁴⁸, meaning prefrontal-thalamic connections in humans remain poorly understood. The data we present here helps overcome these hurdles and demonstrate how synchronised low-frequency activity facilitates interactions between the human cortex and thalamus.”

As for the question about whether the MD-PFC circuit relates to the pulvinar/LGN/primary visual cortex: To our knowledge, there is very little known about such a network. However, after additional analysis suggested by the reviewer in the next point, we have tentative evidence to suggest that such a network does exist, in which the PFC/MD drives visual cortical signals prior to stimulus onset. Further details can be found in our response to the comment immediately below.

As a control for this, perhaps the authors could perform a similar PSI analysis using a searchlight for some of your key metrics across your sensors with >4 or 5 participants, to ensure that we understand the brain network involved? Or, at least, sensors from the P03/P07 and CPz EEG coordinate regions, where from what I can tell there are several participants in supp fig 7

This is a really interesting suggestion – by shifting our focus to the cortex as a whole, we were able to observe a second, posterior region approximating the occipital lobe that also linked to the mediodorsal thalamus. Intriguingly, rather than leading the thalamus (as done by the PFC), this occipital region followed the thalamus, suggesting a network starting at the PFC, moving to the mediodorsal thalamus, and onto the occipital lobe. Given the limited sensor coverage over posterior regions (see supplementary figure 7), however, we would be uncomfortable resting any major conclusions on this pattern of connectivity, but we have described this network in the results (see first quote below; see page 6, lines 123-128), appraised the result in the discussion (see second quote below; see page 9, lines 253-263) and included a new supplementary figure visualising the results (see supplementary figure 10).

“Intriguingly, we also observed directed connectivity in which low-frequency activity in the mediodorsal thalamus preceded low-frequency activity posterior sources (mean cluster $t(5) = -8.15$, $p_{\text{clus}} = 0.063$, $BF_{10} = 73.96$). Given that MEG coverage of these posterior sources was inconsistent across participants (see supplementary figure 7), we have decided to avoid resting any major conclusions based on these thalamus-to-posterior cortex connections. Nonetheless, the interested reader can turn to supplementary figure 10 for more details.”

“Beyond the prefrontal cortex, numerous other cortical regions have been shown to engage in visual perceptual processes (e.g., the dorsal attention network; ^{12,26}). Due to the positioning of the iEEG wires in the MEG, however, we were unable to reliably record signals from these regions, and hence investigate how they interact with the mediodorsal thalamus. Despite this however, we observed interesting connectivity dynamics where the low-frequency thalamic activity seemingly leading low-frequency activity in the occipital cortex (see supplementary figure 10). In the context of the prefrontal connectivity patterns, one could speculate that signals from the prefrontal cortex pass to the occipital lobe via the mediodorsal thalamus, and may explain why phase opposition effects can be seen across the cortex ^{e.g.11,12,26}. Of course, given that these results depend on signals generated from sources with poor MEG sensor coverage, one must take these findings with a grain of salt.”

(N.B.: We elected to do this on the source level rather than sensor level as MEG sensors do not follow the same labelling as EEG, so there is no PO3/PO7/CPz to speak of. Moreover, the positioning of the patients in the MEG was somewhat shifted to the

position of typical participants owing to the intracranial wires, meaning estimating the approximate position of PO3/PO7/CPz for scalp-level analysis is challenging [this shift in positioning is inherently accounted for and corrected in source-level analyses]).

Supplementary Fig. 10. Low-frequency thalamic connectivity across the brain. In the main text, we had focused our iEEG-MEG connectivity analyses on the prefrontal cortex due to difficulties recording posterior sources (see supplementary figure 7). Here, however, we plot the connectivity plots for pre-stimulus low-frequency activity (-800ms to 0ms; 6-14Hz) in full. Given the limited and varying coverage of posterior sources, these effects should be interpreted with caution. **(a)** Inter-site phase clustering across all trials (left: unthresholded; right: thresholded). As reported in the main text, we observed substantial undirected connectivity between the mediodorsal thalamus and medial prefrontal cortex (mean cluster $t(5) = 19.83$, $p_{clus} < 0.001$, $BF_{10} = 2,218.64$). In addition, we saw a second, smaller cluster of connectivity between the ipsilateral occipital lobe and mediodorsal thalamus. When restricting the analysis space to posterior sources [and hence excluding the prefrontal cortex from analysis], this cluster was found to be significant (mean cluster $t(5) = 13.88$, $p_{clus} < 0.001$, $BF_{10} = 551.72$). **(b)** Phase slope index across all trials (left: unthresholded; middle: positive thresholding; right: negative thresholding). A positive phase slope index indicates that the cortex leads the thalamus, while a negative phase slope index indicates that the thalamus leads the cortex. As reported in the main text, we observed significant directed connectivity from the medial prefrontal cortex to the mediodorsal thalamus (mean cluster $t(5) = 5.33$, $p_{clus} < 0.001$, $BF_{10} = 16.73$). In addition, we observed directed connectivity from the mediodorsal thalamus to the occipital/parietal lobes that trended towards significance (mean cluster $t(5) = -8.15$, $p_{clus} = 0.063$, $BF_{10} = 73.96$). **(c)** Inter-site phase clustering for hits relative to misses (left: unthresholded; right: thresholded). While the difference in connectivity between conditions was greatest between the mediodorsal thalamus and medial prefrontal cortex, this was not significant (mean cluster $t(5) = 5.37$, $p_{clus} = 0.188$, $BF_{10} = 17.13$). **(d)** Phase slope index for hits relative to misses (left: unthresholded; middle: positive thresholding; right: negative thresholding). As reported in the main text, we found that directed connectivity from the medial prefrontal cortex to the mediodorsal thalamus was significantly greater for hits relative to misses (mean cluster $t(5) = 8.26$, $p_{clus} < 0.001$, $BF_{10} = 11.71$). Directed connectivity from the mediodorsal thalamus to the occipital lobe did not differ for hits relative to misses (mean cluster $t(5) = -2.80$, $p_{clus} = 0.844$, $BF_{10} = 2.54$).

point 2: I would have liked to see Fig 1C, Fig 2E, and Supp fig 4 and 5 to have included some more regions to develop an account of the network involved in this alpha-rhythmic influence of perception. could the authors create 2 topographic maps: 1 that shows the PBI distribution using Frontal cortex as a seed, and another using MD thalamus as a seed? (e.g. more red regions show higher PBI)

We feel there may be some confusion about the phase bifurcation index (PBI) here: PBI is a local measure and calculated for every MEG source voxel / bipolar pair of thalamic contacts separately, meaning there is no seed region to speak of. We have however included additional source plots to visualise the magnitude of PBI in other regions linked to visual perception and the dorsal attention network (specifically, the frontal eye fields

and intraparietal sulcus). Hopefully this helps elucidate the network demonstrating the pre-stimulus, low-frequency phase bifurcating phenomenon (see supplementary figure 8).

Supplementary Fig. 8. Previous work has demonstrated that the dorsal attention network shows similar phase bifurcation effects as the prefrontal cortex. Here, we explored whether similar effects can be observed in our data, taking two regions that have previously been shown to demonstrate rhythmic sampling: the frontal eye fields (panel a) and the intraparietal sulcus (panel b) [Fiebelkorn, Minsk & Kastner, 2019]. Panel A has been centred on MNI co-ordinates representative of the left and right frontal eye fields (Murd et al., 2020). Panel B has been centred on MNI co-ordinates representative of the intraparietal sulcus (Bray et al., 2013). While we did not observe any substantial phase bifurcation in either area for the patient data, substantial phase bifurcation was observed in the left frontal eye fields for the healthy controls. The absence of an effect in the patient data may be attributable to the fact that the externalised intracranial wires ran over the back of the head, producing large artifacts that reduce the signal-to-noise of data sampled from these regions. Consequently, the absence of an effect in the dorsal attention network of the patients should not be interpreted as an absence of an effect in the population, but simply an idiosyncratic feature of MEG recordings from patients with externalised DBS wires.

point 3: The sentence "Notably, the ideas presented above do not explain the rhythmicity of our observed effect." on page 6 is confusing. If alpha band connectivity is not an important signal in these regions and for this task, why define your metrics using it? There are other signals in EEG to corroborate this claim. This is especially worrying since most of the stronger effects (Fig 1C, 2A, Supp Fig 4, Supp fig 2) are sometimes closer to what most would call theta band. So, again, call it alpha if this is not important?

When referring to "the ideas presented above...", we had intended to mean the idea of a Bayesian observer presented by Rikhye et al. (2018) who had not considered rhythmicity in their framework, rather the more general ideas about low-frequency connectivity put forth by the manuscript. That said, as pointed out by Reviewer 1, predictive coding accounts such as the "Bayesian observer" will inherently produce patterns of low frequency connectivity between the regions involved in the process (Alamia & VanRullen, 2019). So, while Rikhye et al. (2018) did not explicitly relate their framework to rhythmic activity, the "Bayesian observer" would nonetheless be expected to produce rhythmic patterns of connectivity between the thalamus and cortex. We have updated the paragraph of the Bayesian observer accordingly, and have changed the opening of sentence of the next paragraph (i.e., "...the ideas presented above do not explain the rhythmicity of our observed effect.") to acknowledge the fact that the previous ideas do indeed account for rhythmicity (see page 8, lines 192-193).

"An alternative explanation of the rhythmic corticothalamic interaction stems from works investigating interactions between the pulvinar and cortical attentional networks."

As for whether the effects observed here are better labelled as “alpha” or “theta”, we feel this might be down to the particular preferences of individual readers. We completely agree with the reviewer that in some instances, the effects may be better ascribed to theta (3-8Hz; e.g., Figure 1C, 2A) though others certainly are better ascribed to alpha (8-12Hz; Figure 1F, 1G, 2E). Ultimately, the fact that most effects seem to linger around 8Hz means the terms “alpha” and “theta” seem equally valid. More generally, this highlights the growing feeling within the electrophysiological community that traditional frequency bands are no longer fit for purpose. As such, we had decided to avoid interpreting our results as definitively “alpha” or definitively “theta”, and instead present them as exact numerical values or describe them as ‘low frequency’.

One specific thought: The data all hinge on alpha band waveform characteristics, I think, which is slightly confusing to me. What about using raw or bandpass filtered neural signals across a 4-20 Hz range in thalamus and mPFC to classify whether a target was reported as detected or not? This could be done by on a time point by timepoint basis using logistic regression for example, as long as the authors make sure to resample trials to equalize incorrect and correct ones in the training set.

While certainly an interesting idea, we would argue that a classifier based on logistic regression is ill-suited for assessing phase-related differences between conditions. As phase is a circular metric, and logistic regression assumes that the features used are linear, any analysis based will violate fundamental assumptions made by the logistic model. We have visualised this below.

In **(a)**, we have display the phasic differences between two conditions. Though the phase of the two conditions are continuously and diametrically opposed, there are moments when the linear measure of amplitude is identical for both conditions (i.e. when amplitude = 0). Consequently, any model trying to linearly separate the two conditions will regularly fail to do so – this is visualised in **(b)**, where linear separation between the conditions hits zero twice within a single oscillatory cycle despite the conditions sitting at opposing phases. As such, this produces an oscillating pattern of performance at twice the frequency of the resonating frequencies of the two conditions. We can see this very same effect in the real MEG data (see **(c)**); note that as these classifiers require multiple channels, no such analysis can be conducted on the iEEG). We observe that pre-stimulus decoding performance oscillates at ~20Hz, which would be almost double the frequency when PBI was maximal (see figure 1f), an effect attributable to trying to linearly separate to conditions which have phasic differences.

Ideally, the solution to this problem is a multivariate logistic regression classification algorithm developed for circular data, though to our knowledge, no such procedure has been established by the field, and such a procedure would require extensive validation to test its sensitivity and specificity, we feel that developing such an approach is beyond the scope of this paper. Furthermore, given that several studies investigating pre-stimulus correlates of visual detection have succeeded using the univariate approach as we have done (e.g., Busch et al., 2009; Busch & VanRullen, 2010; Fakche et al., 2021; Zazio et al., 2021), we do not feel there is cause for concern that we have not used multivariate measures here.

point 4: "No robust phase bifurcation was observed in additional anterior thalamic recordings ($t(4) = 4.20$, $p_{\text{clus}} > 0.5$, $\text{BF}_{10} = 5.63$; though no difference in PBI was observed between the anterior and mediodorsal thalami: $t(5) = 7.52$, $p_{\text{clus}} = 0.094$ $\text{BF}_{10} = 25.84$; see supplementary figure 2)." pg 3. This is a bit troubling, and note that the bottom panel of supplementary figure two is really showing a difference in the theta range. Can the authors offer more information as to why this is?

Given that the mediodorsal and anterior thalamus are two distinct nuclei, with distinct anatomical connections (e.g., Carlén, 2017; Grodd et al., 2020), and distinct links to cognition (e.g., Li et al., 2022; Phillips et al., 2021; Sweeney-Reed et al., 2021), the fact that the anterior thalamus does not show any link to visual perception is not surprising and has no bearing on the finding that the MD is linked to visual perception. Rather, it highlights the heterogeneity of thalamic function, and demonstrates why one shouldn't generalise results from one thalamic nucleus to another (an idea that overlaps with our earlier comment about generalising results from the pulvinar to the MD). As such, we do not see the absence of phase bifurcation in the ANT as troubling.

Moreover, though not significant, as the reviewer points out, the direction of MD-ANT difference visualised in the bottom panel of supplementary figure 2 is such that MD PBI exceeds ANT PBI, speaking in favour an interpretation that the MD is more intimately linked to visual perception than the ANT. Of course, given that the effect does not reach the threshold for significance, we did not wish to make strong claims about the distinction, but nonetheless, it provides additional evidence to suggest that there is a functional distinction between the MD and ANT during visual perception.

As for the "theta" effect in the figure, while early parts of the effect seem centred on a theta-like 6Hz, the later parts are centred on alpha-like 10Hz. As mentioned above, we have now updated the paper to avoid explicitly labelling these effects as theta/alpha, and instead have labelled them as "low frequency".

When I went to the Methods to better understand this, I read "For statistical analysis, the z-transformed PBI of each patient were pooled together and subjected to a group-level, cluster-based, permutation test (using 64 permutations; i.e., every possible permutation from a sample of six patients [26])." Could the authors better explain the extent of the clusters and the dimensionality of their thalamic and MEG recordings? It was unclear whether this cluster correction was occurring over spatial regions of the brain or over frequencies?

As we used a single pair of contacts (in bipolar montage) per participant for the iEEG PBI analyses, cluster correction was conducted only across time and frequency. For the MEG data, due to limitations of the Fieldtrip function *ft_sourcestatistics*, we first had to average the PBI values across the pre-stimulus time window (providing a single PBI value for every voxel), and then run the cluster analyses across space. We have clarified this in the methods (see page 13, lines 435-445; and page 13, lines 458-466).

"For statistical analysis, we pooled together the z-transformed PBI of each patient and conducted a group-level, cluster-based, permutation test⁶⁷ (using 64 permutations; i.e.,

every possible permutation from a sample of six patients [2⁶]). To aid in the interpretability of the cluster (that is, one cannot state exact when a “significant” cluster arises, only that has arisen in the time-frequency window analysed; see ⁶⁸), we restricted the cluster analysis to the pre-stimulus period (i.e., -800ms to stimulus onset) and to the frequency range where this effect has been observed in previous studies of the cortex (i.e., 6-14Hz; see ⁶⁹ for meta-analysis). Cluster analysis addressed issues of multiple comparisons across time and frequency while the spectrotemporal region of interest ensured spectral/temporal specificity to pre-stimulus low-frequencies. As we only used a single mediodorsal thalamic channel (derived from a bipolar-referenced electrode pair) from each participant for this analysis, there were no multiple comparisons across space.”

“When statistically appraising phase bifurcation in the patient MEG data (n=6), 64 permutations were used once again. As the function `ft_sourcestatistics` cannot conduct cluster analyses across time/frequency while simultaneously conducting analyses across space, we averaged the PBI values across the pre-stimulus window (i.e., -800ms to stimulus onset) and across the frequency range where this effect has been observed in previous studies of the cortex (i.e., 6-14Hz; see ⁶⁹ for meta-analysis), which provided a single PBI value for each voxel of source-reconstructed MEG data. The cluster analysis was then conducted across space on this time/frequency averaged data.”

I also found it slightly difficult to find out how many iEEG contacts where in the thalamus? Could the authors clarify. More generally, The methods are slightly difficult to read at times.

All patients had eight iEEG contacts implanted that targeted the thalamus (four in each hemisphere), although the final positions of some contacts did sit outside the thalamus. We have updated the methods section to more clearly state the number of contacts that were implanted in each patient (see page 10, lines 324-325 and lines 327-328).

“The two thalamic depth electrodes each had four intracranial electrode contacts (platinum–iridium contacts, 1.5 mm wide with 1.5 mm edge-to-edge distance).”

“All patients received bilateral implants, resulting in eight electrode contacts in the thalamic area.”

We have also added a supplementary table that details exactly where each electrode sat within each patient (see supplementary table 1). We refer to this table on page 11, line 340-341 of the main text.

“Full details of electrode positioning can be found in supplementary table 1.”

We have also taken care to clear up the writing in the methods section, principally by switching from the original “passive” voice to a more “active” voice. We have, however, retained the somewhat clunky “first...”, “second...”, “third...” terminology; while this does not make for the most delightful read, it does clearly describe the temporal order of the analysis pipeline and may help others who wish to use similar pipelines.

Smaller things

1) Is the order of the Butterworth filter somewhere? I missed it if so. All I see on page 8 "Second, the recordings were filtered using a 150Hz Butterworth low-pass filter, two Butterworth band-stop filters (to attenuate line noise; 49-51Hz, 99-101Hz), and a 0.5Hz Butterworth high-pass filter".

Thanks for pointing this out. We had neglected to note that we used 6th order filters in our original submission but have now rectified this (see quotes below; see page 11, lines 345-348 and page 11, lines 371-374).

when describing iEEG preprocessing: "...we filtered the recordings using a 150Hz Butterworth low-pass filter (order = 6), two Butterworth band-stop filters (to attenuate line noise; 49-51Hz, 99-101Hz; order = 6), and a 0.5Hz Butterworth high-pass filter (order = 6)"

when describing MEG preprocessing: "...we filtered the recordings using a 150Hz Butterworth low-pass filter (order = 6), two Butterworth band-stop filters (to attenuate line noise; 49-51Hz, 99-101Hz; order = 6), and a 5Hz Butterworth high-pass filter (order = 6)."

2) Fig 1(e) "Patient-specific observed phase-bifurcation (black line) compared to a surrogate distribution (histogram) for individual peak bifurcation frequencies" - P2 seems to have a rather large PBI and low frequency compared to everyone else. Could the authors create the equivalent fig 1C without that participant? - Additionally, could the authors label Fig 1(d) with the subplots labeled by participant?

We have added a new supplementary figure to visualise the PBI measure without participant 2. The plot demonstrates the same narrowband, low-frequency, pre-stimulus increases in PBI shown in figure 1c (see supplementary figure 3). We refer to the supplementary figure in the figure legend of figure 1.

Supplementary Fig. 3. Phase bifurcation in the mediodorsal thalamus after excluding participant 2. As shown in figure 1e, participant 2 produced a larger phase bifurcation index at a lower peak frequency than others. To see if this drives the effect presented in figure 1c, we re-plotted this figure after excluding participant 2. This plot demonstrates the same narrowband, low-frequency, pre-stimulus increase in phase bifurcation seen in figure 1c.

We have also added labels to figure 1d to identify the corresponding participants.

3) Page 6: "was diminished after accounting for pre120 stimulus thalamic activity ($t(5) = 2.26$, $p = 0.031$, $BF_{10} = 3.06$)" This claim cannot be made using that statistical test. Can the authors directly compare the conditions instead?

We think our original phrasing may have caused confusion here. The test used here contrasts the magnitude of pathway c (i.e., the influence prefrontal cortical activity on behaviour when the thalamus is not accounted for) against the magnitude of pathway c' (the influence prefrontal cortical activity on behaviour after accounting for the thalamus). In other words, this analysis is already a contrast of two conditions. We have rephrased this sentence in an attempt to avoid further confusion (see page 6, lines 135-141).

"Moreover, when contrasting the magnitude of pathway c (that is: the direct influence of pre-stimulus prefrontal cortical activity on behavioural performance without accounting for thalamic activity) against pathway c' (i.e., the direct influence of pre-stimulus prefrontal cortical activity on behavioural performance after accounting for thalamic activity), we found evidence to suggest that the direct influence of pre-stimulus prefrontal cortical activity on behavioural performance was diminished after accounting for pre-stimulus thalamic activity ($t(5) = 2.26$, $p = 0.031$, $BF_{10} = 3.06$)."

4) Fig 3B is never referenced?

We have now fixed this, pointing to figure 3B when presenting the statistical analysis of the indirect pathway (see page 6, lines 132-135).

"In this model, the indirect pathway predicted perceptual performance to a degree greater than what would be expected by chance ($t(5) = 3.85$, $p < 0.001$, $BF_{10} = 12.05$; see figure

3b for participant-specific plots of the observed magnitude for the indirect pathway relative to chance).”

5) I would like more discussion of the masking paradigm, why it was employed, and how it relates to point 2

We used backward masking as a means to control the timing of the visibility of the stimulus on screen. Without such a mask, the stimulus will be visible for a variable amount of time due to retinal after-effects, visual processing, and more (Enns & Di Lollo, 2000). Backward masking ensures precise presentation timing because the target is rendered invisible as soon as the mask appears. We have clarified this in the main text (see page 10, lines 310-313)

Following the blank screen, a mask consisting of an overlay of both arrows appeared for 500ms. This mask ensures that the brain perceives the stimulus for the same amount of time across trials, as the presentation of said mask minimises retinal after-effects and post-stimulus visual processing⁵⁹.

6) The authors state that The medial prefrontal cortex leads the mediodorsal thalamus uniquely for hits in Figure 2. Can the fig 3 schematic somehow incorporate this?

We certainly like the idea of having a schematic detailing performance-dependent connectivity. However, we worry that incorporating performance into the schematic of the mediation model in figure 3A may mislead readers about the exact regression models used in the mediation analysis. As such, we have created a fourth figure that provides a visual depiction of the main findings (see below).

Figure 4. Visual depiction of the main findings. Successful detection of a visual stimulus correlates with several neural phenomena: (1) the stimulus being presented at the optimal, low-frequency phase of ongoing medial prefrontal activity (mPFC in purple; hits in red; misses in grey), (2) the stimulus being presented at the optimal, low-frequency phase of ongoing mediodorsal thalamic activity (mediodorsal thalamus in aqua; hits in red; misses in purple), and (3) directed prefrontal-to-thalamic low-frequency connectivity (hits in red; misses [which displayed undirected connectivity] in grey). Critically, the contribution of the prefrontal cortex appears to be mediated by the mediodorsal thalamus.

7) Can the authors characterize how alpha power changes in the MD thalamus and PFC?

We have now updated supplementary figure 4 (see below) to include power spectra for the medial prefrontal cortex.

Supplementary Fig. 4. Spectral power changes in the mediodorsal thalamus and medial prefrontal cortex. Spectral power within the mediodorsal thalamus did not change as a function of perceptual performance (top right; $t(5) = 2.92$, $p_{\text{clus}} = 0.453$, $BF_{10} = 2.83$). Similarly, spectral power within the medial prefrontal cortex did not change as a function of perceptual performance (bottom right; $t(5) = -3.87$, $p_{\text{clus}} = 0.250$, $BF_{10} = 6.11$). Spectral power was computed using the same time-frequency decomposition parameters as those used for the phase bifurcation analyses, and the resulting power spectra for hits and misses were directly contrasted in a cluster-based, permutation dependent-samples t-test.

Reviewer #3:

This is a very interesting study that links thalamic activity in humans (the measurement of which is rare and novel), and its connections with the prefrontal cortex with perception. Using a simple discrimination of a briefly presented then masked visual stimulus, the authors show that, depending on whether the stimulus was correctly identified, the phase of the alpha-band (6–9 Hz) oscillations in the mediodorsal thalamus, as measured in patients with intracranial electrodes implanted, differed approximately 300–800 ms before the onset of the stimulus. The authors describe this "phase bifurcation" also in a broader frequency range in MEG in the prefrontal cortex in both the patients and a control group.

The surprising result is that whether the subject correctly discriminated the visual stimulus did not depend on the alpha phase at the time the stimulus was presented, but rather it depended on the oscillation phase at a substantial time **before** the stimulus was presented.

We would like to thank the reviewer for taking the time to review the manuscript, and for the positive appraisal. We hope our responses and additional analyses have helped clarify the issues raised.

I have a few specific comments:

1. The manuscript is a bit confusing on this point. It seems based on Figure 1d that the oscillation phase was always consistent at time zero, when the stimulus was presented. That is, the oscillations were always in phase with each other whether or not the stimulus was detected. One might expect that the phase would just be random here, but Figure 1d shows only a single phase. Shouldn't the amplitude drop to zero here if the phases were random? Or is the amplitude normalization performed at each time point somehow? It might be helpful to show a distribution of phases at each time point, for each condition. It's not entirely clear, for example, that a particular phase is required some 400 ms before the stimulus, or that the two conditions just tended to differ in phase during this epoch.

We have tried to resolve this confusion on three fronts. Firstly, this confusion may stem from a poor choice of axis labelling on figure 1D, which gave the illusion of phase consistency at time = zero. We had plotted the cosine of the phase angle, rather than raw amplitude. The use of the cosine of the phase angle means no fluctuations in amplitude would be observed as a function of time in this plot. This is why amplitude does not drop to zero at stimulus onset in the figure. We have updated the plot label to avoid further confusion (see figure 1d).

Second, more generally speaking, we feel it is exceptionally difficult to estimate phase when time exactly equals zero, as filtering will inevitably smear so pre-stimulus and post-stimulus activity into the single sample that arises when the stimulus appears on screen. As such, even if one did plot raw amplitude in place of the cosine of the phase angle, one may not expect to see a robust amplitude drop at time = zero simply because it is near-impossible to estimate what is uniquely occurring at time = zero with great precision.

Finally, as for whether "...a particular phase is required some 400 ms before the stimulus, or that the two conditions just tended to differ in phase during this epoch": we feel this is an important question that directly probes the causal role of pre-stimulus oscillatory activity in visual detection. Unfortunately, like many cognitive neuroimaging

experiments, the correlative nature of our analysis prevents us from answering this question, but this could perhaps be addressed in a future study working with patients with deep brain stimulation implants. We have alluded to the promise of such a technique at several stages during the discussion (see page 7-8, lines 187-191; and page 8, lines 205-206).

“While the correlative nature of our data prevents us testing these ideas, future studies which disrupt corticothalamic interactions (e.g., through direct thalamic stimulation) could directly test the causal nature of these hypotheses.”

“Again, future studies may turn to methods such as brain stimulation to directly test the causal nature of this hypothesis.”

2. The manuscript states that the phase bifurcation index (PBI) was "peaking at 8 Hz, 300 ms to stimulus onset). However, in Figure 1c, there seems to be a larger, more significant, cluster of positive PBI at about 7 Hz, 500 ms before stimulus onset.

This difference seems to stem from the fact that the mean PBI is plotted in figure 1C, while the PBI peak is computed based on the maximum t-value. As both metrics are valid ways of determining a peak, we have rephrased the statement to reflect a more general time/frequency window when the effect arose (see page 3, lines 64-66).

“...we observed a positive PBI in the mediodorsal thalamus that was significantly greater than what would be expected by chance (peaking at 7 to 8Hz, 600 to 300ms prior to stimulus onset...”

3. In some subjects, there seems to be a rapid (less than 100 ms) and noticeable change in phase, immediately after the stimulus onset. Is this consistent across subjects?

To address this, we conducted an analysis typically used to identify phase resets (e.g., Jutras et al., 2013). This, however, did not reveal a significant effect, suggesting that while some participants seem to undergo some rapid reorganisation of phase, this was not consistent across participants. We have included these results in the main text (see page 4, lines 72-80; see quote below) as well as a supplementary figure (see supplementary figure 5).

“Notably, the phase of the ongoing low frequency activity of several participants seemed to undergo a rapid shift reminiscent of a phase reset following stimulus onset (see figure 1d). To investigate this, we looked at how spectral power fluctuated as an interaction between time (pre-stimulus vs. post-stimulus) and signal derivation technique (single trial power vs. trial-averaged power). Previous work²⁰ has suggested that an interaction in which trial-averaged post-stimulus power increases relative to pre-stimulus power, but

Supplementary Fig. 5. Phase reset analysis. To test whether the phase of ongoing activity reset following stimulus onset, we computed low-frequency spectral power (6 to 9Hz; in steps of 1Hz) just before stimulus onset (-400 to 0ms; in steps of 25ms) and just after stimulus onset (0 to 400ms; in steps of 25ms) using 6-cycle wavelets. We conducted this spectral decomposition twice: first, on single trials before averaging the result across trials, and second, on the trial-averaged amplitude. If phase resets after stimulus onset, then phase should align across trials after stimulus onset, while will present as an increase in spectral power for trial-averaged post-stimulus activity relative to trial-averaged pre-stimulus activity. In contrast, no change in spectral power will be observed on the single trial level (for further details, see 20). To statistically appraise the effect, we conducted a 2x2 repeated measures ANOVA to probe how spectral power changed as a function of epoch (pre- vs. post-stimulus) and decomposition method (single trial decomposition vs. trial-averaged decomposition). No significant interaction was observed ($F(1, 5) = 1.04, p = 0.355$), suggesting that phase did not reorganise consistently across participants.

single trial power does not, would indicate that the phase of the signal has aligned across trials. However, we observed no such interaction ($F(1, 5) = 1.04, p = 0.355$); see supplementary figure 5), suggesting that phase did not reorganise consistently across participants following stimulus onset.”

4. There seems to be a sizable difference in the PBI in the MEG between the patient and control groups, after the stimulus onset. Specifically the controls show consistently negative PBI (it is not clear how to interpret negative PBI). The positive PBIs also differ somewhat in their timing pre-stimulus (about 400 ms for the patients and 300 ms in controls). Is this significant?

The original description of the phase bifurcation index (PBI; Busch et al., 2009) suggested that a negative post-stimulus PBI may be related to condition-specific differences in stimulus-evoked responses. In line with this idea, we found a significant post-stimulus difference between hits and misses for the healthy controls ($p = 0.043$; see top right panel of the figure below), but no similar effect in patients ($p = 0.481$; see top left panel of figure below). We then went further and subtracted the mean ERP from the single trials before re-computing the PBI; this removed the large negative PBI from the control data while having no substantial influence on the patient data, suggesting the negative PBI was related to condition-specific differences in the evoked response. We have clarified this in the results (see page 4, lines 92-95) and added a supplementary figure (see supp. fig. 6; see below) to demonstrate how ERP subtraction impacts the PBI figures for the patient and control data.

“There was, however, a strong negative PBI following stimulus onset for the healthy controls relative to the patient sample (mean cluster $t(16) = -6.44, p_{clus} < 0.001, BF_{10} = 1558.32$; see figure 1f and 1g). This negative PBI seemed to be driven by the evoked response to the stimulus (see supplementary figure 6).”

Of course, this answer does not explain why there was a difference in evoked response between the two samples. While we have dug deep to uncover the driver of this difference, we have not been able to arrive at a concrete answer. That said, given that this effect is restricted solely to the post-stimulus window, and no post-stimulus effect could retroactively change a pre-stimulus effect, we feel that this open question does not undermine our central results. We have noted this in the sentence immediately following the quote above (see page 4, lines 95-99).

“We were unable to ascertain why the evoked response effect was restricted solely to the healthy controls, but given that this effect is restricted solely to the post-stimulus window, and no post-stimulus effect could retroactively change a pre-stimulus effect, we feel that this open question does not undermine our central results.”

As for “The positive PBIs also differ somewhat in their timing pre-stimulus (about 400 ms for the patients and 300 ms in controls). Is this significant?”: To test whether the positive PBIs differed between the patients and controls, we directly contrasted the effects in an independent samples t-test. No differences were observed (cluster $t(16) = 3.15, p = 0.662, BF_{10} = 7.71$). We have added this to the results (see page 4, lines 90-92).

“While there were minor differences in the timing and spectral profile of the pre-stimulus effects in the patient and control samples, this was not significant (mean cluster $t(16) = 3.15, p_{clus} = 0.662, BF_{10} = 7.71$).”

Supplementary Fig. 6. Influence of event-related potential on phase-bifurcation measures. Differences in the post-stimulus phase bifurcation index (PBI) can be observed between the patient and control MEG recordings, with the latter showing a temporally and spectrally broad negative effect following stimulus onset. PBI is sensitive to condition-specific differences in evoked responses, which present as large negative PBI values (Busch et al., 2009), leading to the speculation that the decrease in the control sample reflects a condition-specific difference in the evoked response. To formally test these ideas, we first tested for condition-specific differences in the evoked response for the two samples using cluster-based permutation tests on the post-stimulus data (0ms to +800ms). While the patient MEG showed no effect ($p = 0.481$; top left panel), a significant difference was observed for the control MEG ($p = 0.043$; top right panel). [Note that the differences in sub-5Hz activity between the samples is driven by the use of heavier filtering in the patient data to attenuate artifacts introduced by the intracranial wires]. The differences in ERPs would explain why the control PBI shows a negative post-stimulus PBI (middle right), while the patients do not (middle left). Indeed, if the trial-average ERP is subtracted from the single-trial data, the negative PBI in the control MEG data disappears (bottom right), while the patient MEG data remains, more-or-less, unchanged (bottom left).

Note that no negative PBI is observed in the mediodorsal thalamus because there was no condition-specific difference in the thalamic evoked response ($p = 0.509$).

5. In Figure 2c, the caption says that the patient's orange lines should be drawn at different lengths, with the length indicating that patient's consistency. But all of the orange lines have the same length.

Thank you for highlighting this inconsistency, we have adapted the figure so that it displays the participant-specific lengths.

6. The peak PBI is around 7 Hz in the thalamus (iEEG) but something like 11 Hz in the prefrontal cortex (MEG). How are the phase-lags and connectivity measures computed between the two methods if the peak frequencies are not the same? I don't see how it

makes sense. If it's not the same frequency, are you sure you are looking at the same phenomenon?

Yes, it is certainly true that there is a difference in the bifurcating frequencies between the iEEG and MEG (see the first comment of reviewer #1 for a similar comment), despite the connectivity metrics being estimated within a frequency band (that is, the same numerical frequency in both regions).

Mathematically speaking, one might expect both inter-site phase clustering (ISPC) and the phase-slope index (PSI) to detect connectivity between neighbouring frequencies. The ISPC utilises phase estimates from wavelet convolution, which introduces some spectral smearing, meaning a 9Hz wavelet that sits between the 11Hz effect seen in the prefrontal cortex and the 7Hz seen in the thalamus would be somewhat sensitive to both signals particularly if there is no pronounced activity at 8Hz itself). Meanwhile, the PSI calculation has a bandwidth of 6Hz, so PSI at intermediary frequencies would also pick up both the slower thalamic signal and the faster prefrontal signal. Taken together, this would explain why we can observe statistically significant connectivity between the two regions despite differences in the peak frequencies of the two regions.

As for a physiological explanation of how coupling can arise between two regions with differing peak frequencies, we would speculate that this relates to travelling waves between weakly-coupled oscillators. Weakly-coupled oscillators are part of a computational account of travelling waves (Ermentrout & Kopell, 1984) (for empirical support in humans: Zhang et al., 2018) which proposes that signals can be relayed from one region to another so long as the oscillator in the transmitting region possess a higher intrinsic frequency than the receiver region. In the case of our data, we could speculate that a travelling wave propagates from the prefrontal cortex (with a higher intrinsic frequency) to the thalamus (with a lower intrinsic frequencies). Indeed, such an idea neatly dovetails with the phase-slope index analyses that demonstrate a directional connection from the prefrontal cortex to the thalamus. Of course, this is a speculative answer which we cannot address empirically here (analyses of travelling waves typically involves the analyses of many electrodes from ECoG grids, as opposed to the small number of electrodes implanted in DBS procedures). Nonetheless, such an explanation presents an interesting avenue of future research regarding corticothalamic interactions, and may provide an answer as to why two regions with differing bifurcating frequencies may nonetheless reflect the same phenomenon.

We have added an additional paragraph into the discussion to discuss both the mathematical and physiological explanations of how prefrontal and thalamic frequencies may interact (see quote below; see page 8, lines 207-227).

“One may be wondering why prefrontal cortical and mediodorsal thalamic phase bifurcation arose at neighbouring, rather than identical, frequencies (~11Hz and ~8Hz respectively). While the spectral smearing incurred through the use of wavelets for our measure of inter-site phase clustering and the 6Hz bandwidth used for the phase-slope index analyses provide a mathematical explanation of connectivity between the two differing frequency bands, it wouldn't explain the physiological underpinnings of such a phenomenon. We speculate, however, that the observed connectivity in conjunction with the mild difference in frequency may relate to travelling waves (e.g. ^{40,41}); more specifically, travelling waves that come about through weakly-coupled oscillators⁴². Models of weakly-coupled oscillators suggest that travelling waves can couple two regions so long as the oscillator of the transmitting region has a higher intrinsic frequency than the oscillator of the receiving region. In the case of the data presented here, we would anticipate that a travelling wave would begin within the prefrontal cortex (given its higher peak phase bifurcation frequency) and propagate to the mediodorsal thalamus. Notably, such an idea neatly ties to the phase-slope index results which demonstrated directed connectivity from the prefrontal cortex to the mediodorsal thalamus. Moreover, this explanation also aligns with the “Bayesian observer” described above, and the travelling waves inherent in such a hypothesis³⁶. Of course, this remains a speculative interpretation of the frequency

differences between the two regions as very little is known about corticothalamic travelling waves. Consequently, such an explanation presents a novel avenue for future research regarding corticothalamic interactions, and may provide an answer as to why two regions with differing bifurcating frequencies may relate to a shared phenomenon.”

Reviewer #4 (Remarks to the Author):

The manuscript by Griffiths et al. investigates the thalamocortical loop while participants perform a visual detection task.

The manuscript is well written and the methods of analysis seem state of the art.

We would like to thank the reviewer for taking the time to review the manuscript, and for the positive appraisal. We hope our responses have helped clarify the issues raised, in particular, the approach to re-referencing.

Major comments:

The authors should elaborate on their re-referencing.

We concur. We have now included a new subheading of the methods that describes our approach to re-referencing (see quote below; see page 11, lines 352-363). The full subheading can be found below:

“Following artifact rejection, we re-referenced the iEEG recordings using a bipolar re-referencing montage to provide a measure of spatially-specific activity within the anterior and mediodorsal thalamic nuclei. All six patients had at least one bipolar-referenced electrode pair within the mediodorsal thalamus, and five of these patients had at least one bipolar-referenced electrode pair within the anterior thalamus. We first identified all bipolar pairs that would feasibly capture mediodorsal/anterior thalamic activity of a given participant, and then selected the pair which produced the cleanest mediodorsal/anterior thalamic evoked response (see supplementary figure 12 for evoked response of the selected pairs). As we used post-stimulus evoked activity as our selection criteria, and our main analyses focused on the pre-stimulus window, we can assume that this selection procedure did not introduce issues of circularity into our main analyses⁶⁵. Full details of bipolar electrode positioning and pairing can be found in Supplementary Table 1.”

Were contacts in the thalamus re-referenced against a contact in white matter? Were adjacent contacts chosen to form bipolar pair?

We elected to use a bipolar referencing montage as this provides us with greater spatial specificity about the source of the signal than what white matter re-referencing can provide (see quote below; page 11, lines 353-355).

“...we re-referenced the iEEG recordings using a bipolar re-referencing montage to provide a measure of spatially-specific activity within the anterior and mediodorsal thalamic nuclei.”

How were pairs chosen for further analysis? Only one pair per participant? Only mediodorsal thalamic nuclei? Were pairs in anterior thalamic nuclei equally accepted? Should this be mentioned in the discussion?

We chose one bipolar pair of electrodes to represent the mediodorsal thalamus of each participant, and another independent bipolar pair of electrodes to represent the anterior thalamus in the five participants who had electrodes within the anterior thalamus. A sixth patient did not have electrodes in the anterior thalamus. For the selection of electrode pairs, we first identified candidate pairs based on anatomy, and then identified the mediodorsal pair and the anterior pair that presented the cleanest electrophysiological signal (based on post-stimulus evoked responses; see supplementary figure 12). We have clarified this in the new subheading on re-referencing (see quote below; see page 11, lines 357-359).

“We first identified all electrode pairs that would feasibly capture mediodorsal/anterior thalamic activity of a given participant, and then selected the pair which produced the cleanest mediodorsal/anterior thalamic evoked response (see supplementary figure 12 for evoked response of the selected pairs).”

We have also added a supplementary table detailing the positioning and bipolar pairing of the intracranial electrodes for the interested reader (see supplementary table 1). All analyses were performed separately on mediodorsal thalamic pairs, or anterior thalamic pairs. Given the addition of these details to the methods, we feel that additional discussion may be somewhat superfluous.

As the authors deem direct recordings of the thalamus and cortex “incredibly rare”, they might want to cite Sarnthein & Jeanmonod 2007 and 2008.

Certainly. We had not intended to suggest that our data were the only instance of such recordings, and are more than happy to point readers to similar examples, including Sarnthein & Jeanmonod’s previous work, as well as a couple of other examples. We have highlighted these papers in the discussion (see quote below; see page 8, lines 230-232).

“To date, studies of these loops in humans are scarce ⁴³, owing to the fact that simultaneous, direct recordings of the specific thalamic nuclei and cortex are rare (see ⁴⁴⁻⁴⁷ for other examples recording from various thalamic nuclei).”

Minor comments:

Six patients (four female, ...

We have now updated the abstract and introduction to acknowledge the sex of the participants (see quotes below; see page 2, lines 23-24, and page 2, lines 50-52).

*“We analysed simultaneously-recorded thalamic iEEG and whole-head MEG in six patients (**four female, two male**...”*

*“...we analysed simultaneously-recorded intracranial electroencephalography (iEEG; targeting the mediodorsal thalamic nuclei) and whole-brain magnetoencephalography (MEG) in six patients (**four female, two male**)...”*

References

- Alamia, A., & VanRullen, R. (2019). Alpha oscillations and traveling waves: Signatures of predictive coding? *PLOS Biology*, 17(10), e3000487. <https://doi.org/10.1371/journal.pbio.3000487>
- Busch, N. A., Dubois, J., & VanRullen, R. (2009). The phase of ongoing EEG oscillations predicts visual perception. *Journal of Neuroscience*, 29(24), 7869–7876. <https://doi.org/10.1523/jneurosci.0113-09.2009>
- Busch, N. A., & VanRullen, R. (2010). Spontaneous EEG oscillations reveal periodic sampling of visual attention. *Proceedings of the National Academy of Sciences of the United States of America*, 107(37), 16048–16053. <https://doi.org/10.1073/pnas.1004801107>
- Carlén, M. (2017). What constitutes the prefrontal cortex? *Science*, 358(6362), 478–482. <https://doi.org/10.1126/science.aan8868>
- Enns, J. T., & Di Lollo, V. (2000). What's new in visual masking? *Trends in Cognitive Sciences*, 4(9), 345–352. [https://doi.org/10.1016/S1364-6613\(00\)01520-5](https://doi.org/10.1016/S1364-6613(00)01520-5)
- Ermentrout, G. B., & Kopell, N. (1984). Frequency Plateaus in a Chain of Weakly Coupled Oscillators, I. *SIAM Journal on Mathematical Analysis*, 15(2), 215–237. <https://doi.org/10.1137/0515019>
- Fakche, C., VanRullen, R., Marque, P., & Dugué, L. (2021). Alpha phase-amplitude tradeoffs predict visual perception. *BioRxiv*, 1–29.
- Fiebelkorn, I. C., Pinsk, M. A., & Kastner, S. (2019). The mediodorsal pulvinar coordinates the macaque fronto-parietal network during rhythmic spatial attention. *Nature Communications*, 10(1). <https://doi.org/10.1038/s41467-018-08151-4>
- Grodd, W., Kumar, V. J., Schüz, A., Lindig, T., & Scheffler, K. (2020). The anterior and medial thalamic nuclei and the human limbic system: Tracing the structural connectivity using diffusion-weighted imaging. *Scientific Reports*, 10(1), 10957. <https://doi.org/10.1038/s41598-020-67770-4>
- Jutras, M. J., Fries, P., & Buffalo, E. A. (2013). Oscillatory activity in the monkey hippocampus during visual exploration and memory formation. *Proceedings of the National Academy of Sciences*, 110(32), 13144–13149. <https://doi.org/10.1073/pnas.1302351110>
- Li, K., Fan, L., Cui, Y., Wei, X., He, Y., Yang, J., Lu, Y., Li, W., Shi, W., Cao, L., Cheng, L., Li, A., You, B., & Jiang, T. (2022). The human mediodorsal thalamus: Organization, connectivity, and function. *NeuroImage*, 249, 118876. <https://doi.org/10.1016/j.neuroimage.2022.118876>
- Muller, L., Chavane, F., Reynolds, J., & Sejnowski, T. J. (2018). Cortical travelling waves: Mechanisms and computational principles. *Nature Reviews Neuroscience*, 19(5), 255–268. <https://doi.org/10.1038/nrn.2018.20>
- Passingham, R. (2009). How good is the macaque monkey model of the human brain? *Current Opinion in Neurobiology*, 19(1), 6–11. <https://doi.org/10.1016/j.conb.2009.01.002>
- Phillips, J. M., Kambi, N. A., Redinbaugh, M. J., Mohanta, S., & Saalman, Y. B. (2021). Disentangling the influences of multiple thalamic nuclei on prefrontal cortex and cognitive control. *Neuroscience & Biobehavioral Reviews*, 128, 487–510. <https://doi.org/10.1016/j.neubiorev.2021.06.042>
- Rikhye, R. V., Wimmer, R. D., & Halassa, M. M. (2018). Toward an Integrative Theory of Thalamic Function. *Annual Review of Neuroscience*, 41(1), 163–183. <https://doi.org/10.1146/annurev-neuro-080317-062144>
- Saalman, Y. B., Pinsk, M. A., Wang, L., Li, X., & Kastner, S. (2012). The pulvinar regulates information transmission between cortical areas based on attention demands. *Science*, 337(6095), 753–756. <https://doi.org/10.1126/science.1223082>
- Sweeney-Reed, C. M., Buentjen, L., Voges, J., Schmitt, F. C., Zaehle, T., Kam, J. W. Y., Kaufmann, J., Heinze, H.-J., Hinrichs, H., Knight, R. T., & Rugg, M. D. (2021). The role of the anterior nuclei of the thalamus in human memory processing. *Neuroscience & Biobehavioral Reviews*, 126, 146–158. <https://doi.org/10.1016/j.neubiorev.2021.02.046>
- Zazio, A., Ruhnau, P., Weisz, N., & Wutz, A. (2021). Pre-stimulus alpha-band power and phase fluctuations originate from different neural sources and exert distinct impact on stimulus-evoked responses. *European Journal of Neuroscience*, ejn.15138. <https://doi.org/10.1111/ejn.15138>
- Zhang, H., Watrous, A. J., Patel, A., & Jacobs, J. (2018). Theta and Alpha Oscillations Are Traveling Waves in the Human Neocortex. *Neuron*, 1–13. <https://doi.org/10.1016/j.neuron.2018.05.019>

REVIEWERS' COMMENTS

Reviewer #1 (Remarks to the Author):

I had only minor points in my previous review. These are all addressed satisfactorily in the revision, adding interesting layers of information to this paper. I have no further points.

Reviewer #2 (Remarks to the Author):

The reviewers have successfully addressed my comments. My only further suggestion is that the additional comments in the discussion addressing the studies novelty in response to my first point are mentioned clearly in the introduction. Specifically, the sentences starting "Our observation of low-frequency connectivity..." as well as the observation that "PFC/MD drives visual cortical signals prior to stimulus onset" could be suggested in the introduction.

The supplementary figures bolster the findings greatly, as well as the inclusion of figure 4.

Finally, I have one more comment about the conclusion "One may be wondering why prefrontal cortical and mediodorsal thalamic phase bifurcation arose at neighbouring, rather than identical, frequencies (~11Hz and ~8Hz respectively)" Is there any evidence of increase in frequency in the thalamus during these periods of coupling? We may expect such "driving" to occur in a system of weakly coupled oscillators. Such a finding may help to bolster this claim, as well as clarify the results.

Reviewer #3 (Remarks to the Author):

The authors have satisfied my concerns. Although they may not have resolved them, they at least discuss them in the revision. I think that this is an interesting enough study that I am willing to overlook some of the unresolved issues.

Reviewer #4 (Remarks to the Author):

The authors have addressed all points adequately.

Minor:

I think it sufficient to mention the sex of the participants in the methods. It can be deleted from the abstract and the introduction.

Reviewer #2

The reviewers have successfully addressed my comments. My only further suggestion is that the additional comments in the discussion addressing the studies novelty in response to my first point are mentioned clearly in the introduction. Specifically, the sentences starting "Our observation of low-frequency connectivity..." as well as the observation that "PFC/MD drives visual cortical signals prior to stimulus onset" could be suggested in the introduction.

We completely agree. We have now added an additional sentence to the introduction highlighting the lack of studies investigating thalamocortical loops in humans (see first quote below; see page 2, lines 50-51), and have clarified that these effects may well arise before stimulus onset (for example: see second quote below; see page 2, lines 48-50).

"While such low-frequency loops have been demonstrated in animals, evidence for similar loops in humans is scarce."

*"One could therefore postulate that these **pre-stimulus** prefrontal low-frequency rhythms reflect connections to mediodorsal thalamus through so-called thalamocortical loops¹⁷⁻²⁰."*

The supplementary figures bolster the findings greatly, as well as the inclusion of figure 4.

Undoubtedly, many thanks for the suggestions.

Finally, I have one more comment about the conclusion "One may be wondering why prefrontal cortical and mediodorsal thalamic phase bifurcation arose at neighbouring, rather than identical, frequencies (~11Hz and ~8Hz respectively)" Is there any evidence of increase in frequency in the thalamus during these periods of coupling? We may expect such "driving" to occur in a system of weakly coupled oscillators. Such a finding may help to bolster this claim, as well as clarify the results.

This is an interesting idea. To test this, we first calculated the phase-locking value between the PFC and mediodorsal thalamus in a jack-knifed manner (i.e., iteratively leaving one trial out from the calculation) and then contrasted these values to the phase-locking value across all trials to estimate phase-locking for individual trials. We then correlated this single-trial phase-locking estimate with the peak frequency within the mediodorsal thalamus power spectrum from the same trial (with the peak being defined by the Matlab function *findpeaks*). However, we didn't observe any profound link between the two variables on either the participant level or the group level (see figure below).

While this result does not directly support the idea of weakly-coupled oscillators, it also does not directly refute it either. We cannot guarantee that every trial contains a travelling wave, and the inclusion of trials that do not contain a travelling wave would likely obscure the proposed correlative effect. Ideally, we would identify trials containing a travelling wave between the PFC and the mediodorsal thalamus, and then run the analysis on this subset of trials. However, given that the current gold standard for travelling wave analysis requires many electrodes positioned on an intracranial grid (rather than the DBS contacts we have available here), such an analysis is not possible. Given the inconclusive nature of this result, we have elected to not include it in the main text, but have discussed how greater electrode coverage may help further address the dynamics of these PFC-MD connections (see page 8, lines 227-230).

Such a hypothesis does, however, present a novel avenue for future research regarding corticothalamic interactions. Future work with more extensive intracranial recordings (a pre-requisite for travelling wave analysis) may provide an answer as to why two regions with differing bifurcating frequencies may relate to a shared phenomenon.

Reviewer #4

I think it sufficient to mention the sex of the participants in the methods. It can be deleted from the abstract and the introduction.

We have removed the reference to the sex of the participants from the abstract and introduction